# Open-channel block of human TRPV6 by polyamine spermine

Arthur Neuberger [1,7], Irina I. Veretenenko [2,3,7], Alexey Shalygin[4,7], Yury A. Trofimov [2,3], Thomas Gudermann [4,5], Vladimir Chubanov [4] ✉, Roman G. Efremov [2,3,6] ✉ & Alexander I. Sobolevsky [1] ✉

Polyamines are organic cations that are present at sub-millimolar concentrations in the cytoplasm and extracellular fluids and serve as versatile modulators of TRP channels, fine-tuning their functions in physiological and pathological contexts, including pain, inflammation and cancer. Despite extensive functional studies, the structural basis by which polyamines regulate TRP channels remains unclear. Here, we combine calcium imaging, electrophysiology, cryo-electron microscopy, mutagenesis and molecular dynamics simulations to study regulation of human TRPV6 by polyamine spermine. Our functional experiments demonstrate voltage-dependent block of TRPV6-mediated currents by spermine. Cryo-electron microscopy reveals that spermine binds in the open pore of TRPV6, extending along the pore axis through the selectivity filter and central cavity. Mutagenesis and molecular dynamics simulations confirm the main binding site of spermine in the selectivity filter and suggest a stepwise molecular mechanism of channel block that includes two more binding sites in the pore transiently occupied by spermine. Our findings enrich the knowledge about TRPV6 regulation by endogenous factors and provide details of the ion channel blocking mechanism that can be explored for inhibition of this channel in disease conditions.

Natural polyamines, like putrescine, spermidine, and spermine, are ubiquitous low-molecular-weight organic cations present at sub-millimolar concentrations in the cytoplasm and extracellular fluids, where they function as crucial regulators of cellular growth and proliferation[1–3]. Importantly, dysregulated polyamine levels have been associated with many forms of human cancer[4–7]. Apart from contributing to other cellular functions, polyamines have been found to act as potent modulators of different classes of ion channels, including BK channels[8], cyclic nucleotide-gated (CNG) channels[9,10], ionotropic glutamate receptors[11–17], voltage-gated sodium channels[18], and inward-rectifier potassium (Kir) channels[19–21]. In addition, the function of

several members of the transient receptor potential (TRP) channel superfamily representing the melastatin[22,23], canonical[24,25], and vanilloid[26–28] subfamilies was found to be regulated by polyamines. However, the structural basis of polyamines' action on TRP channels remains poorly understood.

The constitutively active Ca[2+]-selective TRPV6 channel, a remarkable representative of the vanilloid subfamily of TRP channels, has been defined as a gatekeeper of Ca[2+] transport by epithelial cells of the intestine, placenta, pancreas, and other organs[29–32]. TRPV6 mutations and abnormal expression[33–37] have been linked to a range of human diseases, including transient neonatal hyperparathyroidism,

[1]Department of Biochemistry and Molecular Biophysics, Columbia University, New York, NY, USA. [2]Shemyakin-Ovchinnikov Institute of Bioorganic Chemistry, Russian Academy of Sciences, Moscow, Russia. [3]Research Institute for Systems Biology and Medicine, Moscow, 117246 Moscow, Russia. [4]Walther-Straub Institute of Pharmacology and Toxicology, LMU Munich, Munich, Germany. [5]Comprehensive Pneumology Center, German Center for Lung Research, 81377 Munich, Germany. [6]National Research University Higher School of Economics, 101000 Moscow, Russia. [7]These authors contributed equally: Arthur Neuberger, Irina I. Veretenenko, Alexey Shalygin. ✉e-mail: vladimir.chubanov@lrz.uni-muenchen.de; efremov@nmr.ru; as4005@cumc.columbia.edu

undermineralization and dysplasia of the human skeleton, hypercalciuria, chronic pancreatitis, various reproductive diseases, Pendred syndrome, and Crohn's-like disease[33,38–49]. Moreover, TRPV6 was found to be overexpressed in some of the most severe human cancers, including leukemia, breast, prostate, colon, ovarian, thyroid, and endometrial cancers[7,34–36,50]. Since $Ca^{2+}$ uptake is linked to cell proliferation and cancer progression, TRPV6 was declared an oncochannel and target for cancer therapies[7,51]. Accordingly, there is a pressing need for a better understanding of how endogenous factors regulate TRPV6 channel activity.

In this study, we show that the polyamine spermine blocks human TRPV6 (hTRPV6) channels in a voltage-dependent manner. Using cryo-electron microscopy (cryo-EM), we solve the structure of hTRPV6 in complex with spermine and identify the spermine binding site in the middle of the ion channel pore. Combining structural and functional results with mutagenesis and molecular dynamics (MD) simulations, we provide a mechanistic link between spermine binding and hTRPV6 inhibition.

## Results

### Channel block of human TRPV6 by spermine

To assess the effect of spermine on human TRPV6 (hTRPV6), we performed $Ca^{2+}$ imaging experiments with HEK 293 cells expressing this channel (Fig. 1a–c). Due to constitutive activity of hTRPV6, the increase in extracellular $Ca^{2+}$ concentration from 0.2 to 2 mM resulted in measurable $Ca^{2+}$ uptake, which was blocked by 10 μM cis-22, a potent and selective TRPV6 inhibitor[52] (Fig. 1a). The hTRPV6-mediated $Ca^{2+}$ uptake was also inhibited by extracellular spermine in a concentration-dependent manner (Fig. 1b). However, at high spermine concentrations (>700 μM), we observed an increase in intracellular $Ca^{2+}$ level, independent of hTRPV6 activity. Therefore, the concentration dependence of hTRPV6 inhibition was analyzed within the 0-700 μM range of spermine concentrations (Fig. 1c), yielding the values of half-maximal inhibitory concentration, $IC_{50} = 485 \pm 11$ μM, and the Hill coefficient, $n_{Hill} = 2.6 \pm 0.2$ ($n = 4$).

We also tested the spermine inhibition by performing patch-clamp recordings of whole-cell currents from hTRPV6-expressing HEK 293 cells using a voltage ramp protocol (Fig. 1d). While hTRPV6 is a $Ca^{2+}$-selective ion channel, it conducts monovalent cations in the absence of extracellular calcium[53]. We therefore systematically examined the effects of intracellular and extracellular spermine on hTRPV6-mediated $Na^+$ and $Ca^{2+}$ currents. First, we studied the effect of 1 mM intracellular spermine on $Na^+$ currents during perfusion of TRPV6-expressing cells with a $Ca^{2+}$-free external saline (Fig. 1e–g). We found that in this setting, the addition of intracellular spermine caused a 2.5- and 3-fold reduction of inward and outward monovalent currents, respectively (Fig. 1f, g). Second, we examined whether intracellular spermine affects inward $Ca^{2+}$ currents measured upon exposure of TRPV6-expressing cells to an external solution containing 2 mM $Ca^{2+}$. In these conditions, application of intracellular spermine also caused 4.3- and 2.7-fold reduction of inward and outward currents, respectively (Fig. 1h–j).

Spermine is an abundant intracellular metabolite that can cross the plasma membrane via multiple mechanisms[54–57]. Therefore, we assessed the effect of externally applied 1 mM spermine on $Na^+$ (Supplementary Fig. 1) and $Ca^{2+}$ currents (Supplementary Fig. 2) and found that TRPV6 remains sensitive to spermine, although the inhibitory effect was less pronounced compared to the application of 1 mM intracellular spermine (Fig. 1e-j), supporting the notion that spermine mainly acts from the cytosolic side.

To further characterize the effects of intracellular spermine on hTRPV6, we employed a voltage step protocol (Fig. 1k). We observed that in the absence of 1 mM spermine, application of 20-mV voltage steps in the range of −160 to +100 mV followed by return to a fixed −100 mV potential elicited characteristic inward and outward

monovalent TRPV6 currents (Fig. 1l). Addition of spermine caused a significant suppression of TRPV6 currents by 4.5-fold at −80 mV and 7-fold at +80 mV, and development of pronounced tail currents (Fig. 1m), which were not detectable in the absence of spermine (Fig. 1l). To better visualize this latter effect, we normalized the tail currents to the mean current amplitudes measured at −100 mV before the pre-pulse application (Fig. 1n). We found that spermine induced the tail currents only after the positive pre-pulse voltages, with the strongest effect following the +100-mV pre-pulse. The observed effects of spermine on tail-current appearance are consistent with voltage-dependent relief of spermine block, although they are not uniquely diagnostic of this mechanism, and more complex state-dependent kinetics remain possible[21].

### Cryo-EM structure of human TRPV6 in the presence of spermine

To explore the molecular mechanism of ion channel block, we purified hTRPV6 protein in the presence of 10 mM spermine and subjected it to single-particle cryo-EM. To avoid potential interference between spermine binding and calmodulin (CaM), which causes TRPV6 inactivation, we used a C-terminally truncated hTRPV6 (hTRPV6-CtD) construct that lacks the CaM binding site but otherwise displays wild-type-like function[52,58–60]. Cryo-EM micrographs of hTRPV6-CtD reconstituted into circularized NW30 nanodiscs (cNW30) showed evenly dispersed particles with substantially diverse angular coverage. Data processing revealed a single, distinct particle population that yielded a 4-fold symmetrical 3D map with the overall resolution of 3.48 Å (Supplementary Figs. 3–4, Supplementary Table 1). The map revealed densities for annular lipids but showed no signs of CaM (Fig. 2a, b) and was of sufficient quality (Supplementary Fig. 5) to build residues 28–637 in each one of four hTRPV6 subunits (Fig. 2c).

The resulting structure of hTRPV6 in the presence of spermine (hTRPV6$_{SPM}$) had the same general architecture as structures of open-state TRPV6 solved previously[32,52,58,60–66] (Fig. 2c). The channel is assembled of four subunits with the central ion conducting pore in the middle of the transmembrane domain (TMD). The channel also includes an intracellular 'skirt', a characteristic feature of the vanilloid-subfamily TRP channels, built from ankyrin repeat domains linked by three-stranded β-sheets, N-terminal helices, and C-terminal hooks. The amphipathic TRP helices, a signature of the TRP channel family, connect the TMD to the C-terminal hook, running nearly parallel to the inner leaflet of the membrane and interacting with both the TMD and the skirt. The TMD consists of six transmembrane helices (S1–S6) and a re-entrant pore loop (P-loop) between S5 and S6. The first four transmembrane helices form the S1–S4 domain, which resembles the voltage-sensing domain in voltage-gated ion channels[67]. S5, P-loop, and S6 contribute to the pore domain, which leans against the S1–S4 domain of the adjacent subunit in a domain-swapped manner[64].

In contrast to structures of open-state TRPV6 solved previously[32,52,60–66], the cryo-EM map of the pore domain region of hTRPV6$_{SPM}$ reveals an elongated, sausage-like density in the center that spans the entirety of the selectivity filter and reaches the bottom of the central cavity (Fig. 2b, Supplementary Fig. 5). This density matches the shape of a spermine molecule stretched along the axis of the channel's 4-fold rotational symmetry (Fig. 2d). It is somewhat weaker than density of the surrounding protein, likely due to the dynamic and not very potent nature of spermine binding. Accordingly, the modeled pose of spermine is likely not precise but rather approximate, especially considering that the blocker in the pore has freedom to rotate around the channel axis of symmetry, also causing blurring of the corresponding density in the cryo-EM map. Nevertheless, the identified location of spermine inside the hTRPV6$_{SPM}$ ion channel pore provides a mechanistic explanation for the channel block observed in functional experiments (Fig. 1).

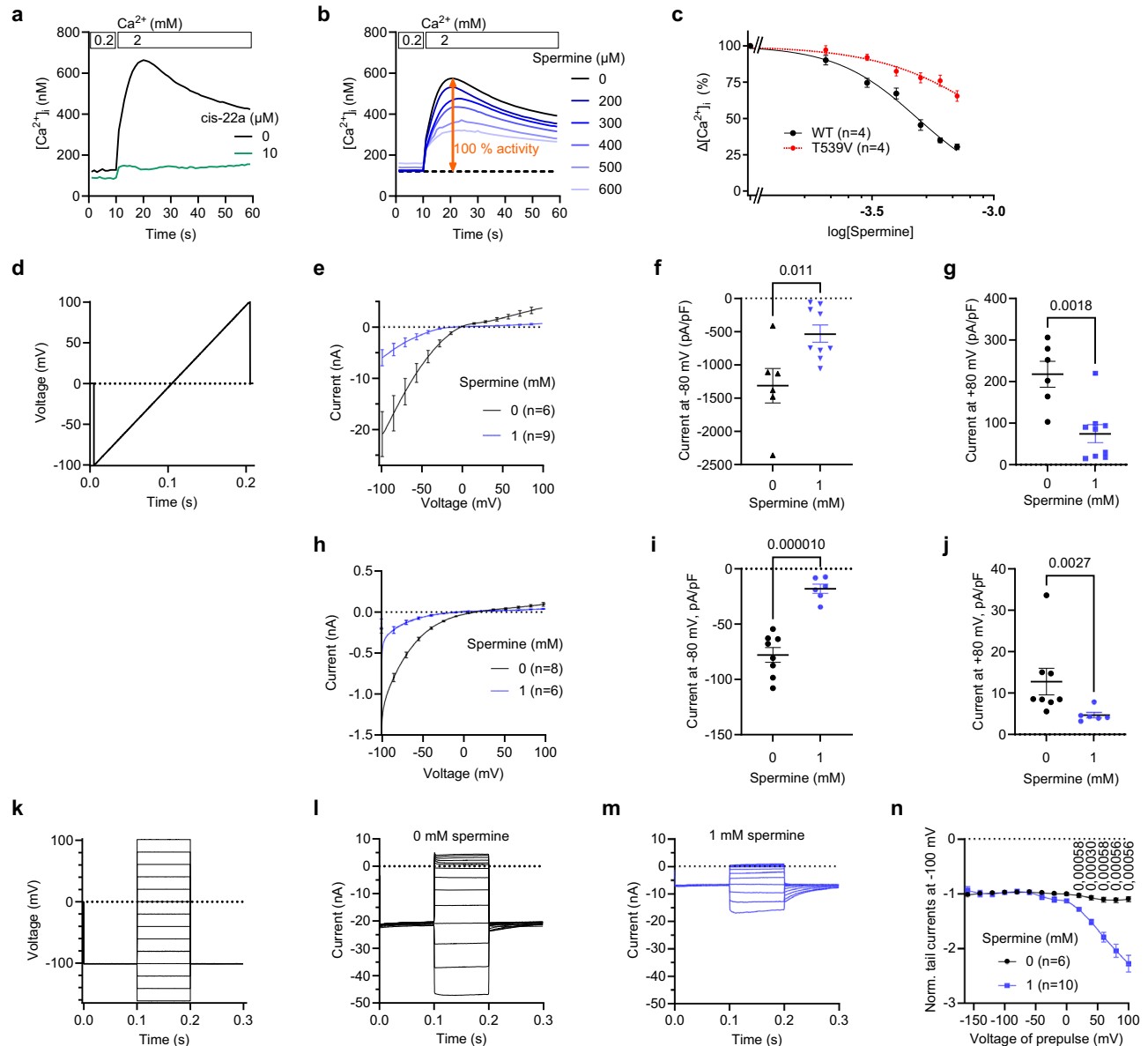

**Fig. 1 | Functional effects of spermine on hTRPV6. a** Representative measurements of $[Ca^{2+}]_i$ in HEK 293 cells expressing hTRPV6-WT exposed to 0.2 or 2 mM extracellular $Ca^{2+}$ in the absence or presence of 10 µM *cis-22*. **b** Measurements were performed analogously to (**a**) but in the presence of indicated concentrations of extracellular spermine. **c** Concentration-dependences for inhibition of hTRPV6-WT and hTRPV6-T539V by extracellular spermine. Data are shown for normalized $\Delta[Ca^{2+}]_i$ calculated from experiments illustrated in (**b**) against the log of the spermine concentration. n is the number of independent experiments. Curves through the points (mean ± SEM) are the logistic Eq. 2 fits; *n* is the number of independent measurements. **d** Voltage ramp protocol. **e** Voltage dependence of whole-cell Na⁺ currents (mean ± SEM) recorded from TRPV6-expressing HEK 293 cells in the absence (black) or presence (blue) of 1 mM intracellular spermine using the ramp protocol shown in (**d**). *n* is the number of cells examined. **f, g** Current amplitudes (mean ± SEM) measured at −80 mV (**f**) or +80 mV (**g**) in the experiment illustrated in (**e**). Data are shown as mean ± SEM; *p* values are shown for the unpaired *t*-test (two-

sided), number of examined cells indicated in (**e**). Source data are provided. **h** Voltage dependence of whole-cell $Ca^{2+}$ currents (mean ± SEM) recorded from TRPV6-expressing HEK 293 cells in the absence (black) or presence (blue) of 1 mM intracellular spermine using the ramp protocol shown in (**d**). *n* is the number of cells examined. **i, j** Current amplitudes measured at −80 mV (**i**) or +80 mV (**j**) in the experiments illustrated in (**h**), *p*-values are shown for two-sided unpaired *t*-test (**i**) and Mann-Whitney test (**j**), number of examined cells indicated in (**h**). Source data are provided. **k** Voltage step protocol. **l, m** Median whole-cell Na⁺ currents recorded in the absence (**l**, black) or presence (**m**, blue) of 1 mM intracellular spermine using the voltage step protocol shown in (**k**). **n** Tail currents measured in (**l, m**) at 201 ms were normalized to the average current at −100 mV (from 0 to 90 ms). Data are shown as mean ± SEM, *p* values are shown if $p \le 0.05$ calculated by the unpaired *t*-test (two-sided) with the Holm-Šídák correction for multiple comparisons. *n* is the number of cells examined.

## Open pore of hTRPV6$_{SPM}$

To infer whether spermine produces a pure open-channel block or causes changes in ion channel conformation, we compared the pores of hTRPV6$_{SPM}$ and hTRPV6 in the open apo state (hTRPV6$_{Open}$). The extracellular part of the hTRPV6$_{SPM}$ pore is a selectivity filter that is lined by extended regions of P-loops contributed by four channel subunits (Fig. 3a). The selectivity filter forms binding sites for different

permeating and blocking cations, as well as ruthenium red[58,60,64,68]. The most critical site for $Ca^{2+}$ permeation and channel block by trivalent ions, like $Gd^{3+}$, is formed by aspartates D542 at the extracellular entry to the selectivity filter[58,64,68]. The central and intracellular parts of the pore are lined by S6 helices. In the middle of the pore is the central cavity, which harbors a binding site for hydrated permeant ions[58,64,68], while the intracellular portion narrows down to form the channel

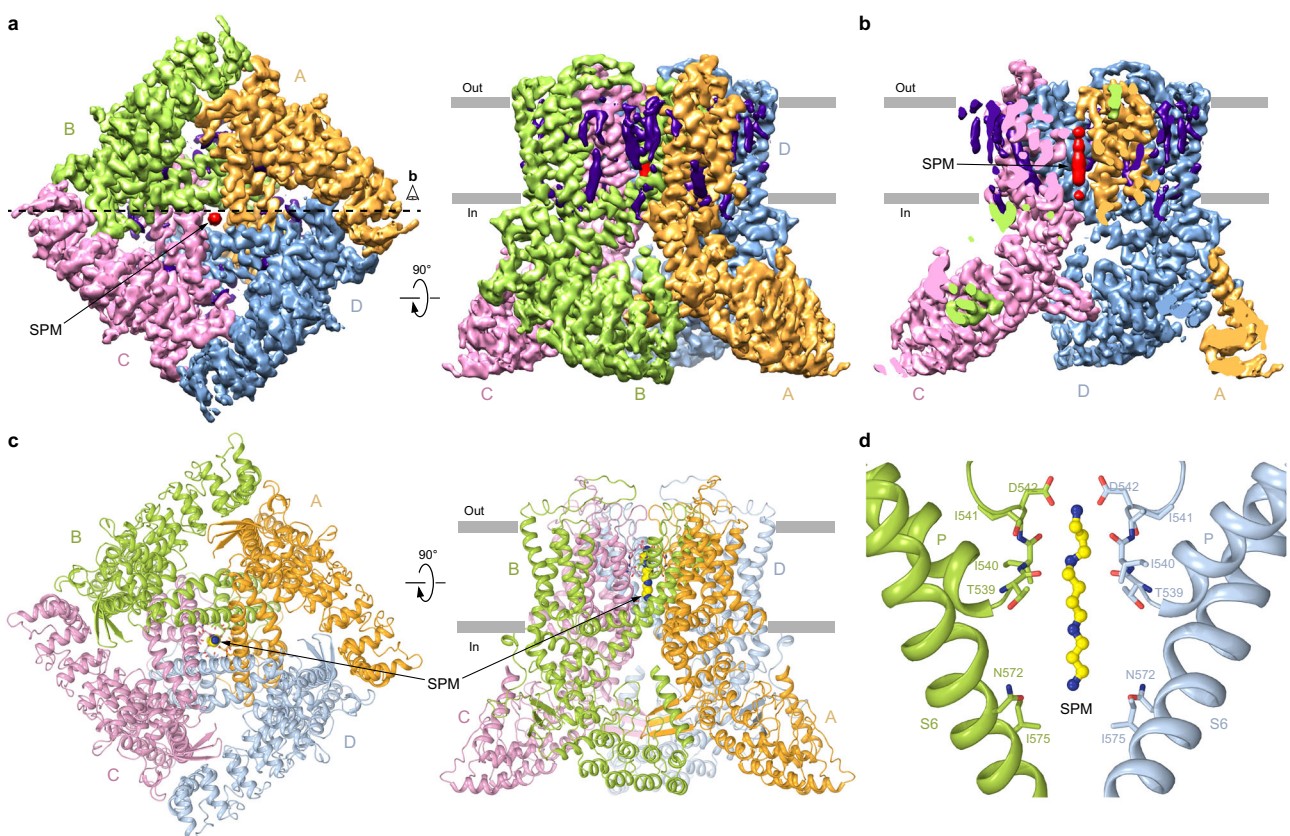

**Fig. 2 | Cryo-EM structure of hTRPV6$_{SPM}$. a** 3D cryo-EM density for hTRPV6$_{SPM}$, viewed intracellularly (left) and parallel to the membrane (right), with TRPV6 subunits colored orange, green, pink, and blue, and density for lipids and SPM colored purple and red, respectively. **b** Density cut off along the dashed line in (**a**). **c** Structural model of hTRPV6$_{SPM}$ viewed intracellularly (left) and parallel to the membrane (right), with coloring of TRPV6 subunits like in (**a**), and SPM molecule shown as a space-filling model. **d** Close-up view of the pore region, with residues contributing to SPM binding shown as sticks. SPM is in ball-and-stick representation. Only two of four subunits are shown, with the front and back subunits omitted for clarity.

gate[58,59,64,65,68,69]. Typical for the open state[52,60–63,65,66], the gate region in the hTRPV6$_{SPM}$ pore is wide, similar to hTRPV6$_{Open}$ (Fig. 3b). Indeed, superposition of the pore-forming regions in hTRPV6$_{SPM}$ and hTRPV6$_{Open}$ (Fig. 3c) as well as measurements of the pore radius (Fig. 3d) clearly demonstrate that the molecular architecture of the hTRPV6$_{SPM}$ pore is nearly identical to that in hTRPV6$_{Open}$. Typical for the TRPV6 open states, the narrowest part of the gate region is formed by side chains of isoleucines I575, while S6 has a π-bulge in the middle, a characteristic feature of the open or inactivated states[59,65]. Hence, hTRPV6$_{SPM}$ represents the ion channel in the open state.

### Molecular dynamics (MD) simulations of spermine blocking of the TRPV6 pore

To get further insights into spermine binding to the human TRPV6 channel, we employed MD simulations. We explored hTRPV6$_{Open}$ embedded in a hydrated lipid membrane and spermine molecules constrained within a cylindrical region of 10 Å radius, allowing their free movement within and near the pore (Fig. 4a). In the first setup, which imitates divalent ion-free conditions of the patch-clamp experiments with intracellular spermine application, five independent replicas containing 150 mM Na$^+$ and Cl$^-$ ions were simulated, with spermine initially positioned 40 Å below the intracellular pore entrance. In four replicas, spermine entered the pore and adopted a cryo-EM–like pose within the selectivity filter (SF) after 200-600 ns of MD simulation (Fig. 4b and Supplementary Fig. 6, Supplementary Movie 1). Three metastable poses can be clearly distinguished throughout the stepwise binding pathway. Pose 1 is located at the intracellular entrance to the pore, where the positively charged groups

of spermine form hydrogen bonds (h-bonds), predominantly with D580 (Fig. 4c, d). At this site, spermine is highly dynamic, frequently switching between two orientations, along the pore and transversal to the pore axis. In Pose 2, spermine is trapped in the central cavity and stretched along the pore axis. Its bottom nitrogen atom is located near the D580 residues but does not form h-bonds with them (Fig. 4e). This pose likely arises due to the energetic barrier formed by cations near SF, thus impeding further spermine penetration. When these cations are knocked off by spermine, the latter adopts Pose 3 inside the selectivity filter and forms h-bonds with D542 and T539 (Fig. 4f, g).

Next, we examined whether MD simulations could capture the impact of Ca$^{2+}$ on the trapping of spermine by TRPV6 using two distinct setups. When Ca$^{2+}$ was placed in the selectivity filter, Na$^+$ spontaneously occupied the cavity between spermine and Ca$^{2+}$, and spermine did not displace either ion in any of the three 1000-ns replicas, instead moving reversibly between bulk water, Pose 1, and Pose 2 (Setup 2, Supplementary Fig. 7a–d). However, when spermine was initiated from Pose 2, and Na$^+$ was excluded from the space between spermine and Ca$^{2+}$, spermine rapidly adopted Pose 3 in all replicas (Setup 3, Supplementary Fig. 7e–h). These results suggest that spermine's trapping in Pose 3 is limited by access to the SF rather than its competition with Ca$^{2+}$.

Finally, we performed MD simulations to compare the behavior of extracellular and intracellular spermine. MD simulations with two intracellular spermines revealed that one spermine occupied Pose 3, whereas the second remained in Pose 1 (Setup 4, Supplementary Fig. 8a–d). When two spermines were initially placed extra- and intracellularly (Setup 5, Supplementary Fig. 8e–j), the simulations revealed

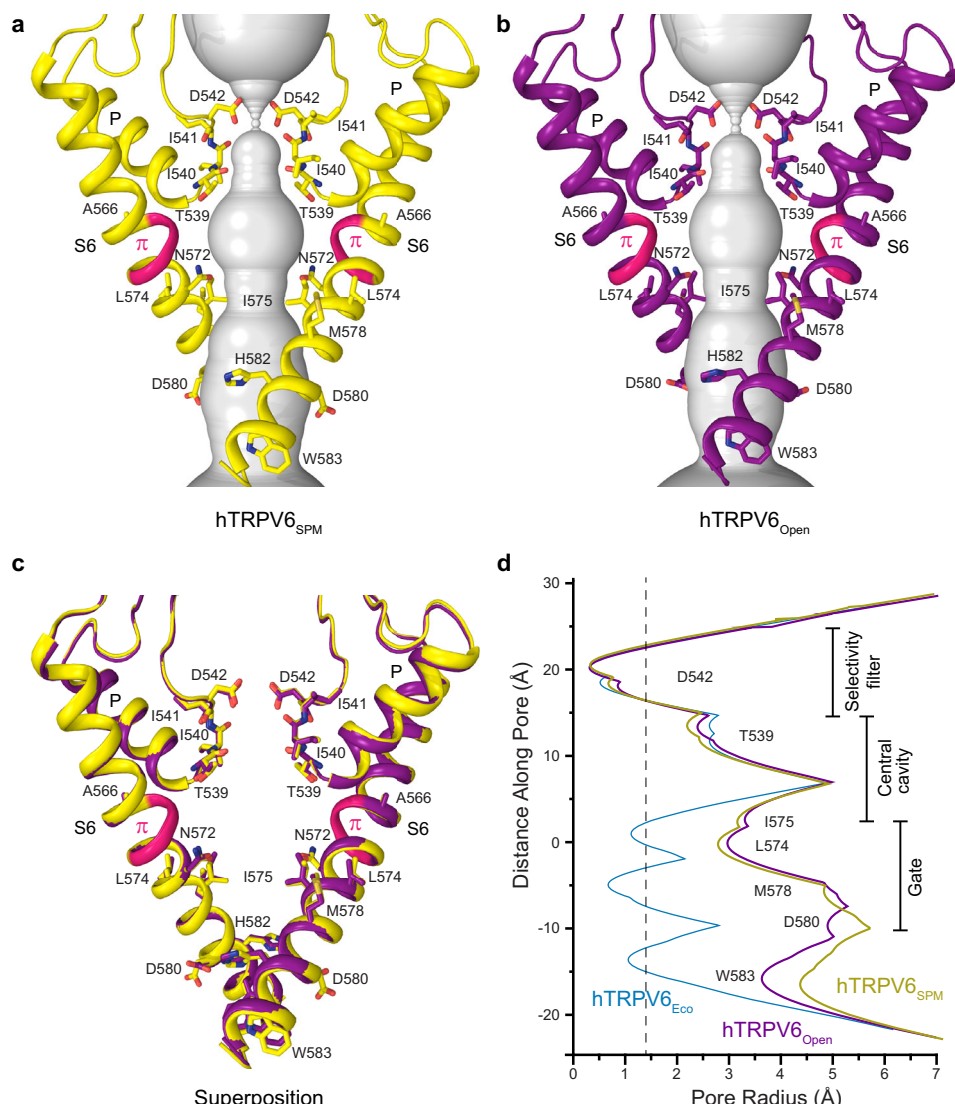

**Fig. 3 | Open pore of hTRPV6$_{SPM}$. a, b** Pore-forming domains in hTRPV6$_{SPM}$ (**a**) and hTRPV6$_{Open}$ (PDB ID: 7S89) (**b**), with residues contributing to pore lining shown as sticks. Only two of four subunits are shown, with the front and back subunits omitted for clarity. The pore profile is shown as a space-filling model (gray). The region that undergoes the α-to-π transition in S6 is highlighted in pink. **c** Superposition of pore-forming domains in hTRPV6$_{SPM}$ and hTRPV6$_{Open}$. **d** Pore radius for hTRPV6$_{SPM}$ (yellow), hTRPV6$_{Open}$ (purple), and the closed-state econazole-bound structure hTRPV6$_{Eco}$ (blue, PDB ID: 7S8C) calculated using HOLE. The vertical dashed line denotes the radius of a water molecule, 1.4 Å.

that the extracellular spermine could reach the SF vestibule but did not enter the filter, instead adopting a highly dynamic vestibular pose (Supplementary Fig. 8k). Meanwhile, intracellular spermine advanced to Pose 3 in two replicas after 300–400 ns. Taken together, these findings suggest that the presence of Ca$^{2+}$ in the SF or spermine in the extracellular vestibule of the SF does not prevent the stepwise binding of the intracellular spermine but rather slows it down due to the occlusion by cations escaping from the pore.

The spermine density averaged over the MD trajectories closely resembles the cryo-EM non-protein density within the pore. Indeed, MD Pose 3 corresponds to the sausage-like zone, which splits into three blobs at a higher density threshold (Fig. 4h). Pose 2 may also contribute to the density observed in cryo-EM, which is difficult to resolve due to its partial overlap with Pose 3. Meanwhile, the more dispersed density at the intracellular entrance overlaps significantly with Pose 1. It is worth noting that the π-cation interactions with W853 can additionally contribute to the spermine stabilization in Pose 1 at experimental conditions, while this type of interactions was not explicitly taken

into account in MD simulations. At the same time, we did not observe any spermine trace in the extracellular vestibule in the cryo-EM data (Supplementary Fig. 8k).

Further analysis revealed that in Pose 3, the negative electrostatic potential on the pore surface, especially near residues D542, strongly "attracts" a positively charged spermine molecule. This results in the most robust electrostatic complementarity across all the Poses (Supplementary Fig. 9). Furthermore, in Pose 3, the hydrophilic regions of the pore induce the most complementary (and, therefore, favorable) hydrophilic-hydrophilic contacts on the spermine surface, as revealed by the molecular hydrophobicity mapping (Supplementary Fig. 10). Taken together, these findings support the model that Pose 3 represents the most energetically favorable spermine position within the hTRPV6 pore.

### T539V and D580R mutations alter the hTRPV6 channel block by spermine

To functionally verify the pore block mechanism suggested by cryo-EM and MD simulations, we introduced point mutations of D542, T539

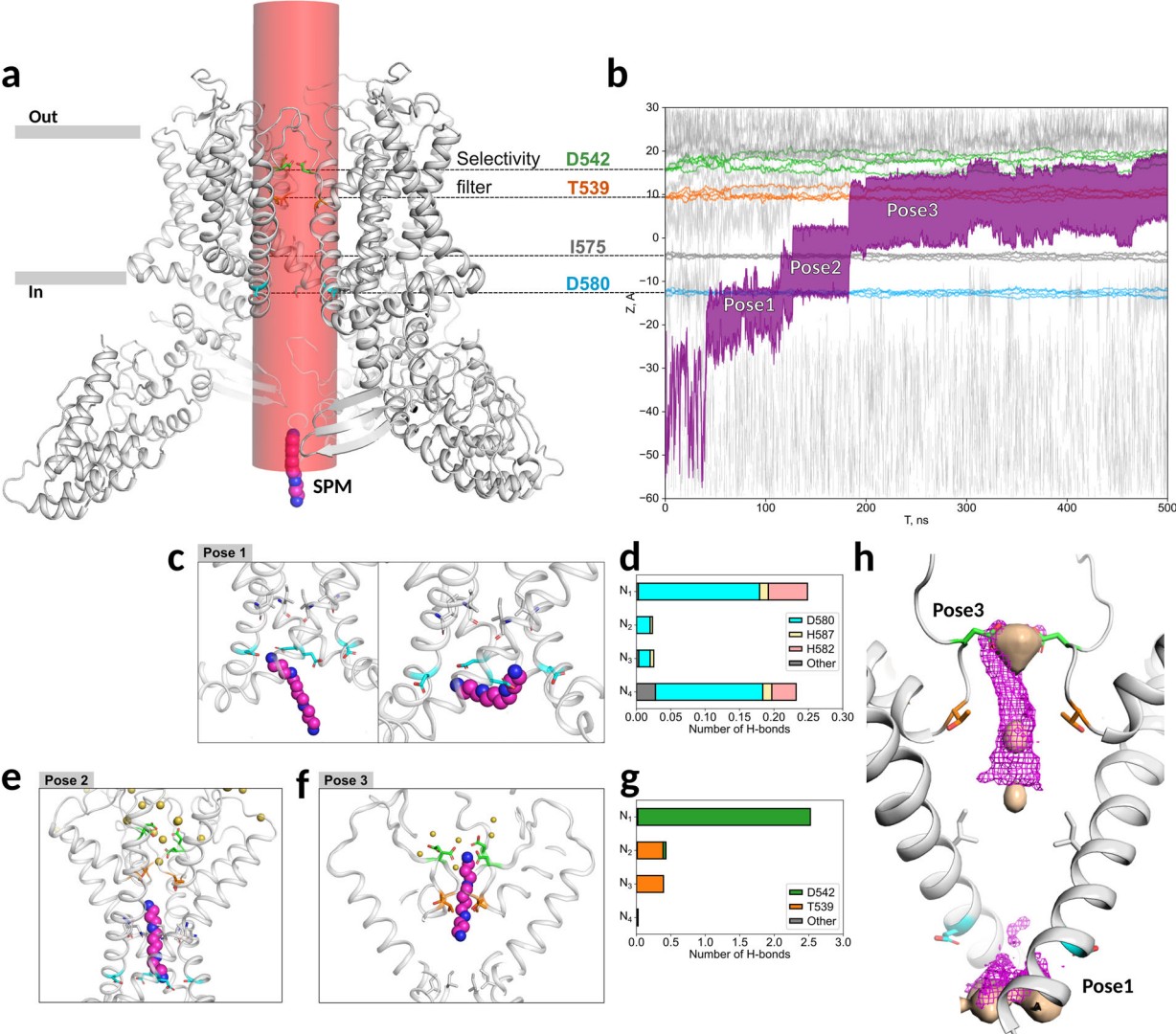

**Fig. 4 | Spermine stepwise binding in the TRPV6 pore revealed by MD simulations. a** First modeling setup: spermine (magenta and blue spheres for C and N atoms, respectively) is initially positioned below the pore entrance in bulk water, and it is free to move within the red cylinder encompassing the pore and its surrounding area. hTRPV6$_{Open}$ is shown in a gray cartoon representation, with gray bars indicating the membrane position. **b** Spermine penetration into the selectivity filter (SF) during one of the simulation replicas. Magenta fillings represent the spermine positions along the pore axis from its top to bottom nitrogen atoms, gray lines show positions of Na$^+$, colored lines correspond to the positions of C$_\alpha$ atoms of D580 (blue), I575 (dark gray), T539 (orange), and D542 (green), dashed lines

project these residues onto the hTRPV6$_{Open}$ structure. Poses 1-3 of spermine are marked on the plot. **c**–**g** Poses 1-3 details: MD snapshots of spermine in Pose 1 in axial (left) and transversal (right) orientations (**c**), in Pose 2 (**e**), and in Pose 3 (**f**). **d, g** average number of h-bonds between the four nitrogen atoms of spermine (N$_1$-N$_4$) and TRPV6 residues in Pose 1 (**d**) and Pose 3 (**g**). **h** MD-averaged spermine densities from the fourth simulation setup (with two spermine molecules) are shown as a magenta mesh overlaid with cryo-EM non-protein densities within the pore (wheat surface). hTRPV6$_{Open}$ is shown in cartoon representation. Positions of Pose 1 and Pose 3 are labeled. In panels (**a**), (**c**), (**e**), (**f**), and (**h**), residues D580, I575, T539, and D542 are shown as blue, gray, orange, and green sticks, respectively.

and D580, which appear to be involved in interactions with spermine (Fig. 5). Consistent with previous studies[70–74], we found that hTRPV6 variants with substitutions of D542 (D542A, D542N, D542E and D542Q) failed to elicit detectable currents in transfected HEK 293 cells as compared to untransfected cells (Supplementary Fig. 11). Therefore, these channel variants were excluded from further analysis. In contrast, we found that the T539 substitution with hydrophobic valine (T539V) did not impair the hTRPV6 channel activity (Fig. 5a, b, d, e). In addition, hTRPV6-T539V retained sensitivity to Gd$^{3+}$ (Supplementary Fig. 12) and econazole (Supplementary Fig. 13). Importantly, Ca$^{2+}$ imaging demonstrated that spermine inhibition of hTRPV6-T539V was much weaker compared to wild-type channels, to the extent that reliable determination of the $IC_{50}$ value was not possible in the experimentally approachable range of spermine concentrations (Fig. 1c).

Next, we used whole-cell patch-clamp recordings and the voltage step protocol to examine how the T539V mutation changes hTRPV6 currents in the absence and presence of 1 mM intracellular spermine (Fig. 5b,e,h). We found that hTRPV6-T539V channels displayed the WT-like current-voltage (I-V) relationship (Fig. 5a, b). However, while the 1-mM intracellular spermine application significantly suppressed the hTRPV6-WT-mediated currents (Fig. 5a), currents through hTRPV6-T539V channels were only weakly inhibited by spermine, and this effect was not even statistically significant at most of the tested membrane potentials (Fig. 5b). Given the notable influence of spermine on hTRPV6 tail currents (Fig. 1k), we explored how the T539V mutation affected this characteristic of hTRPV6 using the voltage step protocol (Fig. 5d, e, g, h). Remarkably, the T539V substitution impaired the development of tail currents (Fig. 5g, h). These results highlight the crucial role of T539 in the ion channel block of hTRPV6 by spermine.

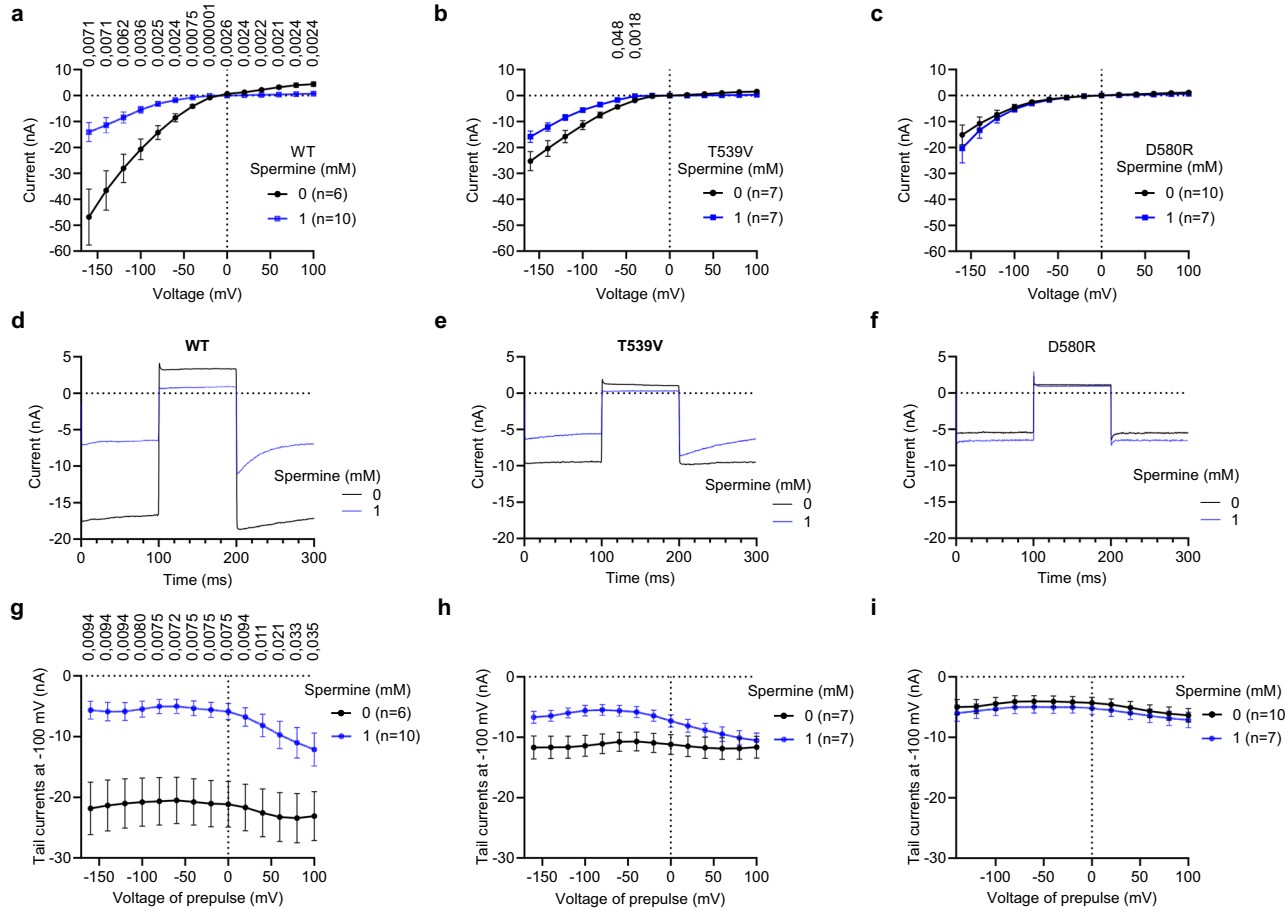

**Fig. 5 | Effect of T539V and D580R mutations on channel block of hTRPV6 by intracellular spermine. a–c** Average voltage dependence of whole-cell currents recorded from HEK 293 cells expressing WT (**a**), T539V (**b**), D580R (**c**) TRPV6 in the absence or presence of 1 mM intracellular spermine using the voltage step protocol shown in Fig. 1 (datapoints for 199 ms of the step protocol). **d–f** Representative currents elicited by the step protocol at +100 and −100 mV for WT (**d**), T539V (**e**), D580R (**f**) TRPV6 with or without 1 mM spermine. **g–i** Initial tail current amplitudes measured for WT (**g**), T539V (**h**), D580R (**i**) TRPV6 at 201 ms after returning to −100 mV during the voltage step protocol with or without intracellular spermine. The data in **a–c**, **g–i** are presented as mean ± SEM, p values are shown for $p \leq 0.05$, calculated by the unpaired t-test (two-sided) with the Holm-Šídák correction for multiple comparisons. n is the number of cells examined.

We also performed MD simulations with hTRPV6-T539V using setup 1 as the initial configuration (with spermine on the intracellular side). In three of the four MD replicas, spermine exhibited the same stepwise penetration into the SF as it did in the WT channel (Supplementary Fig. 14a–d). At the same time, the rate of spermine penetration into Pose 3 was lower compared to WT, and the number of protein-spermine h-bonds was smaller in the same pose (Supplementary Fig. 14e–g), thus suggesting lower spermine binding affinity. These findings are consistent with the less pronounced block of hTRPV6-T539V by spermine compared to WT (Fig. 5).

Finally, using similar electrophysiological settings, we studied whether the exchange of D580 to positively charged arginine (D580R) impacted the inhibitory effect of spermine. We observed that $Na^+$ currents in the cells expressing hTRPV6-D580R were not sensitive to application of 1 mM intracellular spermine (Fig. 5c). In the voltage-step protocol, application of spermine did not alter the characteristics of hTRPV6-D580R currents (Fig. 5f, i). However, hTRPV6-D580R exhibited sensitivity to $Gd^{3+}$ (Supplementary Fig. 12) and econazole (Supplementary Fig. 13), arguing that the D580R mutation did not cause non-specific alterations of hTRPV6 function. These findings support the idea that D580 is critical for the intracellular spermine to enter the ion channel pore of hTRPV6.

MD simulations with hTRPV6-D580R (setup 1) showed that spermine did not penetrate the pore at all in three 1000-ns replicas (Supplementary Fig. 14h–j). This behavior is consistent with an energetic

barrier formed by the positively charged arginines at the intracellular entrance to the pore, which also explains the absence of hTRPV6-D580R inhibition observed in electrophysiological recordings (Fig. 5).

## Discussion

Here, we combined $Ca^{2+}$ imaging, electrophysiology, cryo-electron microscopy, and molecular dynamics simulations to study the regulatory effects of spermine on the human TRPV6 channel. We discovered that spermine enters the channel intracellularly and blocks its pore without altering the channel's open state. According to our model (Fig. 6), the channel pore of hTRPV6 traps the intracellular spermine through three sequential locations referred to as Poses 1-3. Initially, spermine binds to the intracellular entrance of the TRPV6 pore (Pose 1), where it can adopt both axial and transversal orientations. Next, spermine penetrates the pore reaching its central cavity (Pose 2). Finally, spermine moves further and binds to the ion selectivity filter, making polar contacts with side chains of T539 and D542 (Pose 3), consequently occluding permeation through the TRPV6 channel.

Our functional analysis in conjunction with site-directed mutagenesis of T539 and D542 in hTRPV6 aligns well with the proposed model (Fig. 6). By applying alternative approaches, $Ca^{2+}$ imaging and patch-clamp recordings, we showed that spermine inhibits WT hTRPV6 in a concentration- and voltage-dependent manner. Systematic electrophysiological assessment of WT hTRPV6 also revealed that spermine blocks both $Na^+$ and $Ca^{2+}$ currents and that this effect is

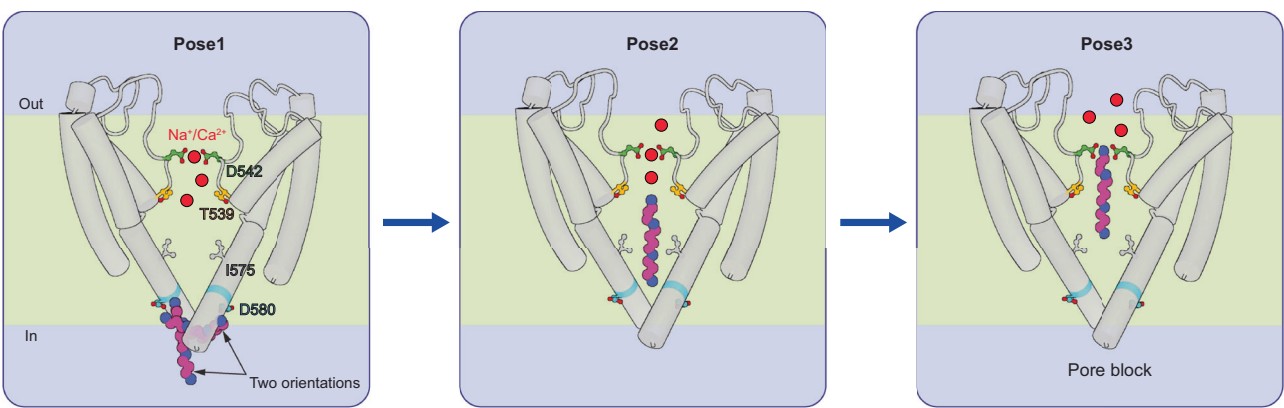

**Fig. 6 | Molecular model of channel block by spermine.** Intracellular spermine initially binds in Pose 1 (left), where it can adopt both axial and transversal orientations, forming polar contacts with the ring of D580 residues. Then, spermine penetrates deeper into the pore, displacing cations and adopting Pose 2 (center), then Pose 3 (right), where it binds the selectivity filter residues D542 and T539, effectively blocking the pore. Spermine is shown as magenta and blue circles, cations – red circles, TRPV6 helices are indicated as gray cylinders, and residues D580, I575 (gate), T539, and D542 are shown as sticks colored blue, gray, orange, and green, respectively.

more pronounced after intracellular application of spermine. Although numerous studies have shown that cells can accumulate extracellularly applied polyamines[75,76], further work is required to establish whether this mechanism solely underlies the effects of external spermine on TRPV6 in our experimental conditions. Voltage-step experiments further revealed that exposure of TRPV6-expressing cells to spermine results in the development of tail currents after positive-voltage pre-pulses, indicating that spermine block is relieved by depolarization. However, more complex effects of spermine on TRPV6 function cannot be fully ruled out. Importantly, mutational analysis indicated that T539 and D580 are critical for spermine block of hTRPV6 currents, supporting our conclusions drawn from the structural and MD analysis.

Interestingly, the pioneering electrophysiological study by Voets et al.[53] examined whether the characteristic outward rectification of TRPV6 depends on intracellular $Mg^{2+}$ or spermine and reported no detectable effects of spermine on TRPV6. However, the inhibitory effects of polyamines were documented for several TRP channels[22–28], including members of the TRPV family, TRPV3 and TRPV4[26,28]. Interestingly, one laboratory reported potentiation of TRPV1 by extracellular polyamines[27], while another group identified channel block of TRPV1 by intracellular spermine[26]. Importantly, despite extensive functional analysis, the structural basis of polyamine interaction with TRP channels remains poorly understood. Thus, a recent cryo-EM reconstruction of TRPV3 in the presence of a synthetic analog of spermine, NASPM (1-naphthyl acetyl spermine), did not reveal density that would correspond to this ligand[28]. Nevertheless, the authors reported conformational rearrangements in the TRPV3 channel in the presence of NASPM and proposed that this molecule suppressed TRPV3 through an allosteric mechanism involving the negatively charged cytoplasmic 'ring' of glutamate residues located prior to the lower gate of TRPV3. We noted that this model aligns well with Pose 1 for spermine in hTRPV6 identified in the present study. An interesting feature of TRPV6 inhibition by spermine is the absence of significant conformational rearrangements in the channel pore. Moreover, even the Pose 1 site, which includes D580 residues, remains stable, although this protein region locally changes its conformation upon binding of another endogenous inhibitor, the magnesium ion[77]. Such differences in the behavior of the TRPV6 pore domain residues indicate its conformational plasticity, which allows the protein to selectively adapt to various ligands and provide the necessary functional response. Further studies are necessary to examine whether the Pose 2 and Pose 3 of spermine in hTRPV6 represent the common polyamine interaction pattern in TRPV3 and other TRP channels.

It is important to emphasize that our study focuses on a particular structure-driven model and, therefore, does not exclude the possibility of alternative mechanisms of hTRPV6 regulation by spermine. As a polyvalent cation, spermine can interact with negatively charged membrane lipids and thereby alter the plasma membrane properties, including surface charge and membrane stability[78]. Such effects could, in turn, modulate the functional properties of TRPV6 through mechanisms distinct from the one proposed here. It is also worth noting that the inhibitory effect of spermine can be complementary to TRPV6 regulation by intracellular $Mg^{2+}$ and CaM[59,66], enabling tight control of TRPV6 constitutive activity, for instance, by preventing $Ca^{2+}$ overload or fine-tuning downstream $Ca^{2+}$ signaling. Since changes in polyamine concentrations were linked to tumor growth[4–7], channel block of TRPV6 by spermine may play a role in cancer progression.

## Methods

### Constructs and cell lines

C-terminally truncated human TRPV6 (hTRPV6-CtD, residues 1–666 of wild-type channel) used in the previous cryo-EM studies of the hTRPV6 channel[52] was cloned into a pEG BacMam vector[79] with a C-terminal thrombin cleavage site followed by a streptavidin affinity tag (WSHPQFEK). Point mutations in wild-type human TRPV6 were introduced using the standard molecular biology techniques as described before[80,81].

For structural experiments, expression of human TRPV6 was performed in HEK 293S cells lacking N-acetyl-glucosaminyltransferase I (GnTI⁻, mycoplasma test negative, ATCC #CRL-3022) that were maintained at 37 °C and 6% $CO_2$ in Freestyle 293 medium (Thermo Fisher Scientific #12338-018) supplemented with 2% FBS (Thermo Fisher Scientific, #10270106). Baculovirus for transducing HEK 293S GnTI⁻ cells were produced in Sf9 cells (GIBCO) cultured in the Sf-900 III SFM media (GIBCO) at 27 °C. For patch-clamp experiments, TRPV6 channels were expressed in HEK 293 T cells (mycoplasma test negative, ATCC #CRL3216) that were maintained at 37 °C and 5% $CO_2$ in DMEM (Merck, #D6429) supplemented with 10% FBS, 100 µg/ml streptomycin and 100 U/ml penicillin (Merck, #P4333).

### Expression and purification

hTRPV6 was expressed and purified using our previously established protocols[52,60–62,66]. Bacmids and baculoviruses were produced using the standard procedures[79,81]. Baculovirus was made in Sf9 cells for ~72 h (Thermo Fisher Scientific, mycoplasma test negative, GIBCO #12659017) and was added to suspension-adapted HEK 293S cells lacking N-acetyl-glucosaminyltransferase I (GnTI⁻, mycoplasma test

negative, ATCC #CRL-3022) that were maintained in Freestyle 293 media (Gibco-Life Technologies #12338-018) supplemented with 2% FBS at 37 °C and 5% CO₂. Twenty-four hours after transduction, 10 mM sodium butyrate was added to the cells to enhance protein expression, and the temperature was reduced to 30 °C. Seventy-two hours after transduction, the cells were harvested by centrifugation at 5471 × $g$ for 15 min using a Sorvall Evolution RC centrifuge (Thermo Fisher Scientific), washed in phosphate-buffered saline pH 8.0, and pelleted by centrifugation at 3202 × $g$ for 10 min using an Eppendorf 5810 centrifuge. The cell pellet was solubilized under constant stirring for 2 h at 4 °C in ice-cold lysis buffer containing 1% (w/v) n-dodecyl β-D-maltoside, 0.1% (w/v) CHS, 20 mM Tris-Cl pH 8.0, 150 mM NaCl, 0.8 μM aprotinin, 4.3 μM leupeptin, 2 μM pepstatin A, 1 mM phenylmethylsulfonyl fluoride, and 1 mM β-mercaptoethanol (βME). The non-solubilized material was pelleted in the Eppendorf 5810 centrifuge at 3202 × $g$ and 4 °C for 10 min. The supernatant was subjected to ultracentrifugation in a Beckman Coulter ultracentrifuge using a Beckman Coulter Type 45Ti rotor at 186,000 × $g$ and 4 °C for 1 h to further clean up the solubilized protein. The supernatant was added to a strep resin and rotated for 14–16 h at 4 °C. The resin was washed with 10 column volumes of the wash buffer containing 20 mM Tris-HCl pH 8.0, 150 mM NaCl, 1 mM βME, 0.01% (w/v) GDN, and 0.001% (w/v) CHS, and the protein was eluted with the same buffer supplemented with 2.5 mM D-desthiobiotin. The eluted protein was concentrated using a 100 kDa NMWL centrifugal filter (MilliporeSigma Amicon) to 0.5 ml and then centrifuged in a Sorvall MTX 150 Micro-Ultracentrifuge (Thermo Fisher Scientific) using an S100AT4 rotor for 30 min at 66,000 × $g$ and 4 °C before injection into a size-exclusion chromatography (SEC) column. hTRPV6 was further purified using a Superose™ 6 10/300 GL SEC column attached to an AKTA FPLC (GE Healthcare) and equilibrated in 150 mM NaCl, 20 mM Tris-HCl pH 8.0, 1 mM βME, 0.01% GDN, and 0.001% CHS. The tetrameric peak fractions were pooled and concentrated using 100-kDa NMWL centrifugal filter to ~3 mg/ml.

hTRPV6 was reconstituted into circularized NW30 nanodiscs (cNW30). cNW30 nanodiscs were prepared according to the standard protocol[82,83] and stored before usage at −80 °C as ~2–3-mg/ml aliquots in the buffer containing 20 mM Tris pH 8.0 and 150 mM NaCl. For nanodisc reconstitution, the purified protein was mixed with cNW30 nanodiscs and soybean lipids (Soy polar extract, Avanti Polar Lipids) at the molar ratio of 1:3:166 (hTRPV6:cNW30:lipids). The lipids were dissolved in the buffer containing 150 mM NaCl and 20 mM Tris, pH 8.0, to reach a concentration of 100 mg/ml and subjected to 5–10 cycles of freezing in liquid nitrogen and thawing in a water bath sonicator. The nanodisc mixture (500 μl) was rocked at room temperature for 1 h. Subsequently, 40 mg of Bio-beads SM2 (Bio-Rad), pre-wet in the buffer containing 20 mM Tris pH 8.0 and 150 mM NaCl, were added to the nanodisc mixture, which was then rotated for one hour at 4 °C. After adding 40 mg more of Bio-beads SM2, the resulting mixture was rotated at 4 °C for another ~14 h. The Bio-beads SM2 were then removed by pipetting. The sample was then centrifuged in a Sorvall MTX 150 Micro-Ultracentrifuge (Thermo Fisher Scientific) using a S100AT4 rotor for 30 min at 66,000 × $g$ and 4 °C before injecting into the SEC column. Nanodisc-reconstituted hTRPV6 was then purified from empty nanodiscs using the Superose™ 6 10/300 GL SEC column equilibrated with the buffer containing 150 mM NaCl, 20 mM Tris pH 8.0, and 1 mM βME. Fractions of nanodisc-reconstituted hTRPV6 were pooled and concentrated to 2.5 mg/ml using a 100-kDa NMWL centrifugal filter. Spermine was added to TRPV6 at 10 mM final concentration and incubated for 1 h at room temperature before freezing the grids.

### Cryo-EM sample preparation and data collection
UltrAuFoil R 1.2/1.3, Au 300 grids were used for plunge-freezing. Prior to sample application, grids were plasma treated in a PELCO easiGlow glow discharge cleaning system (0.39 mBar, 15 mA, "glow" 25 s, "hold"

10 s). A Mark IV Vitrobot (Thermo Fisher Scientific) set to 100% humidity at 4 °C was used to plunge-freeze the grids into liquid ethane after applying 3 μl of protein sample to their gold-coated side using the blot time of 5 s, the blot force of 5, and the wait time of 15 s. The grids were stored in liquid nitrogen before imaging. Images of frozen-hydrated particles of cNW30-reconstituted TRPV6$_{SPM}$ were collected on a Titan Krios TEM operating at 300 kV with a post-column GIF Quantum energy filter of 20 eV and a Gatan K3 Summit DED camera using SerialEM 4.0. 4,487 micrographs were collected in super-resolution mode with an image pixel size of 0.413 Å across a defocus range of −0.8 to −2.0 μm. The total dose of ~60 e⁻ Å⁻² was attained by using a dose rate of ~16 e⁻ pixel⁻¹ s⁻¹ across 50 frames for a 2.0-s total exposure time.

### Image processing and 3D reconstruction
Data were processed in CryoSPARC 3.3[84] (Fig. S1). 4,487 movie frames were collected and subsequently aligned using the patch motion correction. Contrast transfer function (CTF) estimation was performed on non-dose-weighted micrographs using the patch CTF estimation. Subsequent data processing was done on dose-weighted micrographs. Following CTF estimation, micrographs were manually inspected and those with outliers in defocus values, ice thickness, and astigmatism as well as micrographs with lower predicted CTF-correlated resolution (higher than 5 Å) were excluded from further processing (individually assessed for each parameter relative to the overall distribution). We first performed template picking with a previously published open-state map of hTRPV6 (PDB ID: 7S89) with resulting picked particles being extracted first at 4x bin and then sorted in 2 rounds of 2D classification. 507,766 of particles selected in the second 2D round were then further sorted in several consecutive rounds of 3D classifications, first with 4x binned and later with unbinned particles using intermediate good-quality maps that were refined (homogeneous and non-uniform refinements) as templates for the classifications. The reported resolution of 3.48 Å of the final maps following homogeneous and non-uniform refinement (Table S1) was estimated using the gold standard Fourier shell correlation (GSFSC). The local resolution was calculated with the resolution range estimated using the FSC = 0.143 criterion. Cryo-EM density visualization was done in UCSF Chimera 1.17[85] and UCSF ChimeraX 1.5[86].

### Model building
The hTRPV6$_{SPM}$ structural model was built in Coot 0.9.8.1[87], using the previously published cryo-EM structure of hTRPV6 in the open state (PDB ID: 7S89)[60] as a guide. The resulting model was real space refined in Phenix 1.19.2[88] and visualized in UCSF Chimera, UCSF ChimeraX, and PyMOL 2.5.2[89]. The pore radius was calculated using HOLE 2.1[90].

### Patch-clamp experiments
Patch-clamp experiments were performed as reported previously with a few modifications[66]. HEK 293T cells (mycoplasma test negative, ATCC #CRL3216) were maintained at 37 °C and 5% CO2 in DMEM (Merck, #D6429) supplemented with 10% fetal calf serum (Thermo Fisher Scientific, #10270106), 100 μg/ml streptomycin and 100 U/ml penicillin (Merck, #P4333). For transient transfection, cells were grown in 35-mm dishes to ~60% confluence. Using Lipofectamine 2000 (Thermo Fisher Scientific), cells were transfected with 0.5 μg human TRPV6 cDNA[79]. 18–22 h after transfection, whole-cell currents were measured in GFP-positive cells using an EPC10 patch-clamp amplifier and PatchMaster software (Version V2x92, Harvard Bioscience). Holding potential was 0 mV. Currents were elicited using voltage ramps from −100 to +100 mV over 200 ms, applied every 2 s. The capacitance was measured using the automated capacitance cancellation function of EPC10. The resistance compensation function of EPC10 was used. The extracellular solution contained 140 mM NaCl, 2.8 mM KCl, 10 mM EDTA, 10 mM HEPES, and 10 μM NS8593 (Tocris,

4597) (to block endogenous TRPM7 currents). The intracellular solution included 140 mM NaCl, 2.8 mM KCl, 10 mM EDTA, and 10 mM HEPES. For Ca²⁺ influx patch clamp recordings, the extracellular solution contained 140 mM NaCl, 2.8 mM KCl, 2 mM CaCl₂, 1 mM MgCl₂, 10 mM HEPES-NaOH, and 11 mM glucose, and 10 μM NS8593, whereas the intracellular pipette solution contained 120 mM Cs-glutamate, 8 mM NaCl, 10 mM Cs-EDTA, and 10 mM HEPES-CsOH. Solutions were adjusted to a pH of 7.2 using an FE20 pH meter (Mettler Toledo) and an osmolarity of 290 mOsm using a Vapro 5520 osmometer (Wescor). Alternatively, TRPV6 currents were elicited using the step protocol, comprising a holding potential of 0 mV, 100-ms prepulse at −100 mV, followed by 100-ms steps from −160 to +100 mV (20 mV increments) and completed with a 100-ms pulse at −100 mV. In both approaches, the effect of spermine was examined by adding 1 mM spermine (Merck, 85590). Data are shown as mean ± SEM. Unless stated otherwise, the results were analysed using the unpaired t-test with the Holm-Šídák correction for multiple comparisons, as implemented in GraphPad Prism 10.5.0. Significance was accepted at p ≤ 0.05.

**Aequorin-based Ca²⁺ influx assay**
Measurements of intracellular Ca²⁺ concentrations ([Ca²⁺]ᵢ) in TRPV6-expressing cells were performed as reported previously[91,92], with several modifications. Cells cultured in 6-well plates (~60% confluence) were transfected with 1 μg/dish human TRPV6 plasmid DNA and 0.1 μg/dish pG5A plasmid DNA encoding EGFP fused to *Aequorea Victoria* aequorin using Lipofectamine 2000 (Thermo Fisher Scientific, #11668019). Twenty-four hours after transfection, the cells were washed with HEPES-buffered saline (HBS) containing 150 mM NaCl, 5.4 mM KCl, 0.2 mM CaCl₂, 1 mM MgCl₂, 5 mM HEPES (pH 7.4), and 10 mM glucose, and mechanically resuspended in HBS. For reconstitution of aequorin, cell suspensions were incubated with 5 μg/ml coelenterazine (Carl Roth, #4094.3) in the HBS for 30 min at room temperature. Cells were washed twice by centrifugation at 2000 rpm for 5 min (Heraeus Pico 17 microcentrifuge, Thermo Fisher Scientific), resuspended in HBS, and aliquoted into 96-well plates (1×10⁵ cells per well). Luminescence was detected at room temperature using a CLARIOstar microplate reader (BMG LABTECH GmbH). To monitor TRPV6-mediated Ca²⁺ influx, the extracellular concentration of Ca²⁺ ([Ca²⁺]ₒ) was increased to 2 mM by injecting the CaCl₂-containing HBS. The experiments were terminated by lysing cells with 0.05% (v/v) Triton X-100 in HBS to record the total bioluminescence. The bioluminescence rates (counts/sec) were analyzed at 1-s intervals and calibrated as [Ca²⁺]ᵢ values using the following equation:

$$p \times [Ca2+]i = 0.332588 \times (-\log(k)) + 5.5593 \quad (1)$$

where k represents the rate of aequorin consumption, i.e., counts/s divided by the total number of counts.

The concentration dependencies for spermine were fitted using GraphPad Prism 10.5.0 and the following equation:

$$Y = 100/(1 + 10^{((\text{LogIC}_{50} - X)^{*}n_{\text{Hill}})}) \quad (2)$$

where Y is the normalized response, X is the log of spermine concentration, and $n_{\text{Hill}}$ is the Hill coefficient.

**MD simulations**
Structural models of the full-length hTRPV6_Open (PDB ID 9CUJ)[66] were inserted into a hydrated lipid bilayer consisting of palmitoyloleoyl-phosphatidylcholine (POPC) lipids (c.a. 600 lipids, about 160 × 160 × 120 Å³ - simulation box size). Na⁺ and Cl⁻ ions were added to achieve physiological salt concentration (150 mM) and electroneutrality. To prevent the gate closure, intersubunit constraints were imposed on the distances between Cα atoms of 572, 575, 578 residues. The distances were constrained between each residue of one subunit

with all nine residues of the other three subunits with the force constant of 10 kJ/(mol × Å²). Five simulation setups were implemented (Table S2): (1) initially spermine is positioned in the bulk water, 40 Å below the intracellular pore entrance; (2) the same spermine position and Ca²⁺ embedded in the SF; (3) spermine in Pose 2 and Ca²⁺ in the SF; (4) two spermine molecules in the bulk water, 40 Å below the intracellular pore entrance; and (5) two spermine molecules in the bulk water, the first is 40 Å below the intracellular pore entrance and the second is 40 Å above the extracellular vestibule of the SF. To accelerate simulations, the configuration space, where spermine was far from the pore, was prohibited by confining the spermine molecule within a cylindrical region around the pore axis, allowing free movement only within the pore or in bulk water in the pore vicinity (Fig. 4a). The cylindrical constraints were provided by applying a flat-bottom potential with the radius of 10 Å and the force constant of 10³ kJ/(mol × Å²) on the center of mass of the spermine heavy atoms. Three to five replicas were prepared for each setup. The hTRPV6-T539V and hTRPV6-D580R mutants were produced by introducing T539V and D580R point mutations to the hTRPV6_Open model. The simulation systems with the mutants were then prepared similarly to Setup 1.

All replicas were first equilibrated in several stages: 5 × 10⁴ steps of steepest descent minimization followed by heating from 5 to 310 K during a 200-ps MD-run, then 10 ns of MD run with fixed positions of the protein atoms, 10 ns of MD with fixed positions of the protein backbone, 50 ns of MD with fixed positions of the protein Cα atoms to permit membrane relaxation. Spermine heavy atoms were restrained during all equilibration steps. The force constant k = 10 kJ/(mol × Å²) was used for all atoms restrained. Then, production MD runs of 500–1500 ns were carried out for each replica (Table S2). No transmembrane voltage was applied in any setup. MD simulations were performed using GROMACS 2024.4 package[93], CHARMM36m force field[94] with NBFIX corrections[95], and the TIP3P water model[96]. Simulations were carried out with an integration time of 2 fs, constrained hydrogen-containing bond lengths by the LINCS algorithm[97], imposed 3D periodic boundary conditions, constant temperature (310 K) maintained by the v-scale thermostat[98] and constant semi-isotropic pressure (1 bar) maintained by the Parrinello-Rahman barostat[99]. Cut-off distance of 12 Å was used for evaluation of nonbonded interactions and the particle-mesh Ewald method[100] was employed for treatment of long-range electrostatics. Multi-site Ca²⁺ model (CAM) was used for calcium ions[101], which is optimized for the modeling of Ca²⁺-protein interactions[66,69,102]. CHARMM General Force Field (CGenFF)[103,104] was used to generate the spermine topology with a charge of +4e.

H-bonds between spermine and TRPV6 were determined according to the commonly used geometric criteria: a bond was present between the donor (D) and acceptor (A) groups if the distance between them was less than 3.5 Å and the angle D-H-A was 180 ± 30°. The Molecular Surface Topography (MST) tool[105] available at https://model.nmr.ru/cell, was used to map and visualize the electrostatic potential (ESP, Supplementary Fig. 9) and molecular hydrophobicity potential (MHP, Supplementary Fig. 10) on the TRPV6 pore and spermine surfaces. The ESP was calculated according to the atomic partial charges of the protein and spermine for a dielectric constant of ε = 1. The MHP approach enables the quantitative estimation of the spatial distribution of hydrophobic/hydrophilic properties on a molecular surface. The formalism of MHP assumes that each atom in the molecular system is assigned a specific hydrophobicity constant, and the sum of the atomic contributions is calculated at the molecular surface[105]. In this study, we used the atomic hydrophobicity constants obtained by Wildman and Crippen[106]. Such an approach provides a useful tool to estimate the complementarity of intermolecular interactions: contacts between oppositely charged molecular surfaces, as well as hydrophobic-hydrophobic and hydrophilic-hydrophilic contacts, have high complementarity, while other contacts are unfavorable (Trofimov et al.[105] for further details). MD data were visualized in Pymol[89].

**Reporting summary**

Further information on research design is available in the Nature Portfolio Reporting Summary linked to this article.

## Data availability

The cryo-EM density map of hTRPV6 in complex with spermine was deposited to the Electron Microscopy Data Bank (EMDB) under the accession code EMD-76261. The atomic coordinates have been deposited to the Protein Data Bank (PDB) under the accession code 12AG. All MD simulation setups and coordinates of the protein and spermine obtained in MD simulations can be found via the Github platform [https://github.com/Gressy2113/TRPV6-SPM.git] and on Zenodo as entry 19729146. Source data are provided with this paper.

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

## Acknowledgements
We thank Joanna Zaisserer and Anna Erbacher (Walther-Straub Institute, LMU Munich) for their technical assistance. Access to computational resources of HPC facilities at NRU HSE, the Supercomputer Center "Polytechnical" at the St. Petersburg Polytechnic University, and IACP FEB RAS Shared Resource Center "Far Eastern Computing Resource" equipment (https://cc.dvo.ru) is gratefully appreciated.

## Author contributions
A.N. carried out protein expression, protein purification, cryo-EM sample preparation, and cryo-EM data processing. A.S., T.G., and V.C. performed $Ca^{2+}$ imaging and electrophysiological experiments. I.I.V., Y.A.T., and R.G.E. performed MD simulations and MD analysis. A.N. and A.S. made constructs. A.N. and A.I.S. analyzed structural data and built molecular models. A.N., I.I.V., A.S., V.C., and A.I.S. wrote the manuscript, which was then edited by all authors.

## Funding
A portion of this research was supported by NIH grant U24GM129547 and performed at the PNCC at OHSU and accessed through EMSL (grid. 436923.9), a DOE Office of Science User Facility sponsored by the Office of Biological and Environmental Research. As a Walter Benjamin Fellow, A.N. was funded by the Deutsche Forschungsgemeinschaft (DFG, German Research Foundation) – 464295817. I.I.V., Y.A.T., and R.G.E. were supported by the MSHE Agreement no. 075-15-2024-536. T.G. and V.C. were supported by DFG TRR 152 (P15) and GRK 2338 RTG (P10). A.I.S. was supported by the Human Frontier Science Program (HFSP) Award and the NIH (AR078814, CA206573, NS083660, NS107253, HL181985).

## Competing interests
The authors declare no competing interests.
