## [Transparent Peer Review file · Nature Communications]

Open-channel block of human TRPV6 by polyamine spermine

Corresponding Author: Dr Alexander Sobolevsky

Version 0:

Reviewer comments:

Reviewer #1

(Remarks to the Author)

The manuscript by Neuberger et al. presents a structural and functional characterization of the interaction between the polyamine spermine and the human TRPV6 channel. Given the biomedical relevance of TRPV6 as an “oncochannel” and the upregulation of polyamines in cancer, this study addresses a timely and significant question regarding endogenous channel regulation. The authors report that spermine acts as a voltage-dependent channel blocker, a finding supported by electrophysiology and cryo-EM data showing an electron density within the pore. With this, a multi-step binding mechanism is proposed, supported by molecular dynamics simulations.

Overall, the combination of cryo-EM, electrophysiology, and MD provides a comprehensive narrative. However, there are critical discrepancies with historical literature (specifically regarding the novelty of the blocking effect) and interpretative leaps in the structural data that require rigorous address before acceptance.

General Comments

A 20 year old work on TRPV6 biophysics, specifically Voets et al. 2003 (<https://doi.org/10.1085/jgp.20028752>), explicitly reported that TRPV6 rectification was not altered by intracellular concentrations of spermine. The authors must discuss this direct contradiction. Is it a difference in ionic conditions, voltage protocols, or expression systems? A direct side-by-side comparison or discussion is necessary.

2. Interpretation of Cryo-EM Density The resolution of the reconstruction is 3.48 Å. While sufficient for backbone tracing, assigning orientation and identity to a flexible linear molecule like spermine (“sausage-like density”), based solely on this resolution is perhaps questionable. The density in Figure 2 appears continuous from the selectivity filter (SF) down to the central cavity. Spermine is approximately 15-20 Å long. Does the length of the density strictly match a single spermine molecule, or could it represent a mixture of states or even lipids/detergent? The MD simulation suggests dynamic “Poses,” but the EM map shows a single extended density. The authors should clarify/discuss if the density represents an average of multiple spermine molecules or a single stable occupancy. This contrasts with the recent TRPV3-NASPM study (Zhang et al., 2025, cited) where no pore density was found at similar resolution. The authors correctly highlight this difference, but they should do a better job in convincing the readers that the density in TRPV6 isn’t an artifact of the nanodisc/conditions.

3. The reported Hill coefficient is 2.6 ± 0.2 . This somewhat suggests cooperativity. The structural model (Fig 2) and MD (Fig 6) largely focus on a single spermine molecule traversing the pore axis. How does the binding of one blocker to the central pore axis explain a Hill coefficient of ~ 2.6 ? Does this imply multiple spermine molecules bind simultaneously (as hinted at in MD Setup 5, Fig S5), or is it an allosteric effect involving the four subunits? The current “Pose 1-3” narrative (sequential single-molecule entry) does not fully explain the cooperative functional data. Please expand.

Specific Comments

4. Functional Characterization in figure 1 (Ca²⁺ Imaging): The inhibition curve looks robust. However, the increase in intracellular Ca²⁺ at >700 μM spermine is attributed to “targeting of other Ca²⁺ transporting proteins.” This sounds very speculative please elaborate the idea better.

5. Tail Currents on figure 1: The tail currents recordings are very enlightening. The text states they “reach the maximum at +100 mV.” On one hand, the phrasing might be confusing for broad audience.; tail currents are usually measured at a fixed return potential after a prepulse. On the other hand, the figure shows them increasing with the prepulse voltage. This could be an indication of a relief from blocking or slow unblock kinetics. The total abolition of this phenotype in T539V (Fig 5) is a strong piece of evidence and should be discussed more in the main text as proof of the specific interaction site.

6. Pose Stability in MD simulations: The MD data is used to bridge the gap between the static structure and function. "Pose 2" is described as transient. If it is transient, why does the EM density appear continuous through that region? If the EM density is an average, it implies high occupancy in both the SF and the central cavity simultaneously? please discuss.

7. The proposed mechanism is "Stepwise." Pose 1 (D580) to Pose 2 to Pose 3 (SF). The mutagenesis of D580R supports Pose 1 as an entry site. However, if D580R abolishes block, does it prevent entry or binding? The currents in D580R look large (Fig 5c), so the channel is functional. This could represent a control that validates the MD. For the D580R mutant, the authors show it is insensitive to spermine. It would be valuable to see if this mutant retains sensitivity to other blockers (like Ruthenium Red or cis-22) to prove the pore structure isn't radically perturbed, only the polyamine entry path.

Reviewer #2

(Remarks to the Author)

Summary:

The study by Neuberger et al. describes the cryo-EM structure of human TRPV6 captured in the presence of the polyamine spermine. Polyamines are ubiquitous and play critical roles in numerous essential life processes. They are dysregulated in many cancer types and, of particular relevance to this manuscript, broadly impact the function of numerous ion channels. How polyamines modulate the electrical character of these ion channels is unknown. In functional studies, the authors observe dose-dependent block of TRPV6 when spermine is administered from both the extracellular or intracellular side. Interestingly, intracellular spermine produced voltage-dependent tail currents in the presence of spermine, consistent with an open-channel block mechanism. The authors present a structure of human TRPV6 with spermine lodged in the selectivity filter (SF) of the channel pore. This is the major finding of the paper. MD simulations further spermine interacting with three intrapore locations, with one of the sites overlapping with the spermine-binding site observed in the cryo-EM structures. Mutagenesis and functional analysis confirm the importance of these sites for spermine-dependent channel modulation. Together, these findings support an open-channel block model in which spermine enters the pore from the cytosol to obstruct ion flow in a use-dependent manner.

Major Comments:

1. While the overall cryo-EM reconstruction of the channel at 3.48 Å appears sound and in line with current standards, the evidence of the ligand pose is substantially weaker than for the surrounding protein scaffold. The cryo-EM density authors attribute to spermine is not well-defined, appearing fragmented and located on the four-fold symmetry axis, which raises concerns of over-interpretation. Spermine, as presented, is modelled asymmetrically with an internal "kink", despite being situated in the symmetry axis of the SF and is not convincingly supported by the density or by clear specific interactions with surrounding residues. Given the somewhat fragmented density at this site, such fine-grained conformational detail is unlikely to be reliably resolved. The manuscript could benefit from more rigorous quantitative validation of the spermine model and from tempering statements that imply an unambiguous binding mode.
2. The MD simulations demonstrate that spermine docks at three intrapore poses and with greatest stability at the SF (pose 3) close to the position taken up by spermine in the cryo-EM structure obtained in the absence of Ca²⁺. While this correlation is intriguing, this does not by itself establish that this is the predominant functional blocking site. Additionally, simulations with Ca²⁺ present indicate that Ca²⁺ occupancy of the SF prevents spermine docking to the SF from the intracellular side. This finding appears to argue against pose 3 as the dominant blocking configuration under physiological conditions where Ca²⁺ is expected to permeate the channel. This important constraint is not discussed. Furthermore, it is unclear whether Ca²⁺ flow can be blocked if spermine were to occupy only poses 1 and/or 2. It would strengthen the manuscript to clearly distinguish between what is directly supported by the structural and MD data (i.e. a plausible binding region and plausible blocking configurations under specific access and ionic conditions) versus what is hypothetical. The latter should be explicitly framed as a working model rather than as a definitive mechanism.
3. The MD simulations are used as key support for the proposed inhibition mechanism but the scope is more restricted than the text implies. While the simulations demonstrate the stability of intracellular spermine docking to pose 3 in the SF under strong constraints, they do not establish that spermine can access the pore from the extracellular side or that the full inhibition pathway has been captured. The manuscript does not clearly identify which pose is proposed to mediate block under physiological Ca²⁺ with extracellular spermine versus intracellular spermine in Ca²⁺-free conditions.
4. The functional evidence for block is distributed across assays that probe different conditions and access route. Calcium imaging demonstrates reduced Ca²⁺ influx when spermine is applied extracellularly under physiological Ca²⁺, whereas whole-cell recordings are performed in nominally Ca²⁺-free conditions with spermine in the pipette solution. These regimes are not equivalent and make it difficult to extract a single, unified blocking mechanism. With the electrophysiological data, the appearance of prominent, voltage-dependent tail currents only in the presence of the ligand, despite stable, non-inactivating currents during the depolarizing prepulse and minimal tails in the control, suggests that spermine not only blocks the pore but also alters deactivation gating and/or stabilizes an additional kinetic state. The current text focuses on voltage-dependent open-channel block and does not discuss the likely impact on deactivation kinetics or the state-dependence of block under the different ionic and spermine-application conditions. A more balanced interpretation that explicitly considers spermine-induced changes in deactivation would provide a more accurate structure-function picture and prevent over-simplification of the mechanism.

Reviewer #3

(Remarks to the Author)

The manuscript by Neuberger et al. reports on an interesting investigation of the inhibition mechanism of TRPV6 by spermidine. The authors use a combination of techniques including cryoEM, molecular dynamics, and electrophysiology to

clarify the molecular underpinnings of the regulation of human TRPV6 by polyamine spermine. The paper sheds light on one of the fundamental mechanisms of modulation of ion channels and provides novel structural information about the binding of polyamines to TRPV6. The results are of sufficiently wide interest to warrant publication. I find that the simulations are rigorously carried out and that the manuscript is, for the most part, clearly written and easy to read. I recommend publication after the following concerns have been addressed:

1) While I find the information provided by the molecular dynamics simulations insightful, one is left wondering whether different replicas would end up resulting in the same poses as those described in the paper. I think that a potential of mean force for the binding/unbinding of the polyamine would make the arguments discussed in the paper more solid and reliable.

2) The calculations used to produce the plots in figs S6 and S7 (ESP and MHP) should be described in the methods. At the moment, the manuscript refers to a webserver, which, in turn, cites the relevant paper. A concise summary of the methodology should be reported in the manuscript.

Version 1:

Reviewer comments:

Reviewer #1

(Remarks to the Author)

The authors handled my critiques well.

Reviewer #2

(Remarks to the Author)

I appreciate the revision and the additional experiments. The manuscript is improved and the new data strengthen the conclusion that spermine inhibits TRPV6. My remaining concerns are mainly about mechanistic interpretation.

1. My original comment on the tail-current phenotype appears to have been misread. The phrase "stable, non-inactivating currents during the depolarizing prepulse and minimal tails in the control" referred to the control condition, not to spermine. My point was that the ligand-dependent emergence of strong tail currents introduces kinetic complexity that deserves balanced discussion.

2. I remain unconvinced that the tail currents uniquely support a simple interpretation of relief of spermine block at positive membrane potentials. Although the tails are consistent with voltage-dependent changes in spermine-channel interactions, the authors' own response emphasizes that the prepulse currents in the presence of spermine remain strongly suppressed at positive potentials. The current text should be toned down to state that the phenotype is consistent with, but not uniquely diagnostic of voltage-dependent relief of block and that more complex state-dependent kinetics are not excluded.

3. The explanation offered for extracellular spermine effects, namely uptake into cells and indirect access to the intracellular blocking pool, remains speculative in the context of the present study. If retained, this point should be either supported directly or clearly identified as speculative. The new external patch-clamp data are helpful but do not fully resolve this issue.

4. Patch clamp experiments establishing the concentration dependence of intracellular spermine block would still help interpret the calcium imaging and electrophysiology results

Reviewer #3

(Remarks to the Author)

My concerns have been addressed. Even though PMFs would strengthen the work, they're not crucial to support the conclusions. I recommend publication

We thank all Reviewers for their excellent suggestions that have led to significant improvement of this manuscript. We have made changes throughout the manuscript with the details outlined in our responses below.

Reviewer #1 (Remarks to the Author):

The manuscript by Neuberger et al. presents a structural and functional characterization of the interaction between the polyamine spermine and the human TRPV6 channel. Given the biomedical relevance of TRPV6 as an “oncochannel” and the upregulation of polyamines in cancer, this study addresses a timely and significant question regarding endogenous channel regulation. The authors report that spermine acts as a voltage-dependent channel blocker, a finding supported by electrophysiology and cryo-EM data showing an electron density within the pore. With this, a multi-step binding mechanism is proposed, supported by molecular dynamics simulations.

Overall, the combination of cryo-EM, electrophysiology, and MD provides a comprehensive narrative.

We thank Reviewer #1 for the positive and generous assessment of our work.

However, there are critical discrepancies with historical literature (specifically regarding the novelty of the blocking effect) and interpretative leaps in the structural data that require rigorous address before acceptance.

General Comments

A 20 year old work on TRPV6 biophysics, specifically Voets et al. 2003 (<https://doi.org/10.1085/jgp.20028752>), explicitly reported that TRPV6 rectification was not altered by intracellular concentrations of spermine. The authors must discuss this direct contradiction. Is it a difference in ionic conditions, voltage protocols, or expression systems? A direct side-by-side comparison or discussion is necessary.

We thank Reviewer 1 for pointing out this work. We have discussed this study in the revised manuscript (lines 296-298).

The manuscript by Voets et al. primarily focused on testing the hypothesis that the inward rectification of TRPV6 depends on intracellular Mg^{2+} . A side-by-side comparison of these results with our experiments was already reported in our recent publication (doi: 10.1038/s41467-025-65919-1). Briefly, our study recapitulates the functional impact of intracellular Mg^{2+} on TRPV6 currents observed by Voets et al. In addition, we provided the structural basis of direct Mg^{2+} interaction with TRPV6.

Unfortunately, the manuscript by Voets et al. does not provide sufficient methodological details or primary data regarding their experiments with spermine. The only information we could identify is a brief statement in the Results section (page 251): ***“To investigate whether a similar mechanism causes rectification of TRPV6 in the absence of intracellular Mg^{2+} , we first measured whole-cell currents using a Mg^{2+} -free intracellular solution containing 5 mM spermine. Currents measured under these conditions were time-independent, displayed moderate inward rectification (rectification score = $10.1 \pm$***

0.7; n = 4) and were indiscernible from currents measured in the absence of intracellular spermine and Mg²⁺ (unpublished data).” Given the limited methodological description and the absence of actual experimental data, a side-by-side comparison of the results reported by Voets et al. and those presented in our manuscript is problematic. We can only note that in our hands, TRPV6 retains inward rectification in the absence of spermine (Fig. 1), and in this respect, our results do not contradict the idea of Voets et al. that TRPV6’s inward rectification is not underscored by intracellular spermine.

2. Interpretation of Cryo-EM Density The resolution of the reconstruction is 3.48 Å. While sufficient for backbone tracing, assigning orientation and identity to a flexible linear molecule like spermine (“sausage-like density”), based solely on this resolution is perhaps questionable. The density in Figure 2 appears continuous from the selectivity filter (SF) down to the central cavity. Spermine is approximately 15-20 Å long. Does the length of the density strictly match a single spermine molecule, or could it represent a mixture of states or even lipids/detergent? The MD simulation suggests dynamic “Poses,” but the EM map shows a single extended density. The authors should clarify/discuss if the density represents an average of multiple spermine molecules or a single stable occupancy. This contrasts with the recent TRPV3-NASPM study (Zhang et al., 2025, cited) where no pore density was found at similar resolution. The authors correctly highlight this difference, but they should do a better job in convincing the readers that the density in TRPV6 isn’t an artifact of the nanodisc/conditions.

Under the same experimental conditions, but in the absence of spermine, we did not observe a comparable density in the pore of TRPV6, strongly suggesting that the observed density corresponds to spermine. Moreover, this interpretation is supported by MD simulations and by mutagenesis combined with functional recordings. At the same time, we agree with Reviewer #1 that the observed density may represent multiple spermine poses within the pore, as also indicated by the MD simulations. We have now made this point explicit in the revised manuscript (lines 211-219). TRPV3 has distinct functional properties and differs substantially from TRPV6 in its amino acid sequence. Therefore, we are cautious about drawing conclusions from the absence of spermine molecules in the recently resolved TRPV3 structure when interpreting our results.

3. The reported Hill coefficient is 2.6 ± 0.2 . This somewhat suggests cooperativity. The structural model (Fig 2) and MD (Fig 6) largely focus on a single spermine molecule traversing the pore axis. How does the binding of one blocker to the central pore axis explain a Hill coefficient of ~ 2.6 ? Does this imply multiple spermine molecules bind simultaneously (as hinted at in MD Setup 5, Fig S5), or is it an allosteric effect involving the four subunits? The current “Pose 1-3” narrative (sequential single-molecule entry) does not fully explain the cooperative functional data. Please expand.

It is important to note that the concentration-response relationship for spermine was obtained from Ca²⁺ imaging experiments following extracellular application of the ligand. Under these conditions, the apparent Hill coefficient may reflect processes beyond pore binding itself, including membrane permeation and intracellular accumulation of spermine. Therefore, the Hill coefficient cannot be reliably interpreted to support a specific mechanistic model of TRPV6 inhibition. Instead, IC₅₀ values and Hill coefficients were determined

empirically to provide an appropriate quantitative description of the inhibitory effect of spermine on TRPV6. Interestingly, our MD analysis indicates that two spermine molecules can occupy the TRPV6 pore simultaneously. To this end, a cooperative effect at high spermine concentrations on TRPV6 cannot be ruled out. However, as explained above, it would be premature to link MD-based results with the 2.6 Hill coefficient calculated in Ca^{2+} imaging experiments.

4. Functional Characterization in figure 1 (Ca^{2+} Imaging): The inhibition curve looks robust. However, the increase in intracellular Ca^{2+} at $>700 \mu\text{M}$ spermine is attributed to “targeting of other Ca^{2+} transporting proteins.” This sounds very speculative please elaborate the idea better.

Thank you for pointing out this issue. We observed that application of spermine at concentrations 0.8-1 mM to the Ca^{2+} -impermeable TRPV6 variant (D542A-TRPV6) caused a significant increase in intracellular Ca^{2+} levels. **Fig. R1** below shows an example of our experiments with the D542A-TRPV6 variant examined in the absence and presence of 1 mM spermine. We observed that treatment with spermine increased Ca^{2+} levels by ~2-fold under resting conditions (in the presence of 0.2 mM Ca^{2+}), indicating that high levels of spermine can alter cytosolic Ca^{2+} levels in HEK293 cells independently of TRPV6. Spermine is known to modulate several Ca^{2+} -related targets, including NMDA, AMPA, and kainate receptors, as well as IP3 receptors, ryanodine receptors, and TRPC channels; spermine has also been implicated in CaSR-dependent signalling involving Orai1/TRPC1-mediated Ca^{2+} entry. PMID: 1825128; 15574796; 39592599; 8280134; 1322698; 26631167; 29926068. Through these interactions, spermine may promote Ca^{2+} influx or Ca^{2+} release from intracellular stores, thereby increasing cytosolic Ca^{2+} . As these mechanisms are outside the scope of the present study, we did not perform further experiments to elucidate the nature of this phenomenon in our settings. However, we edited the text (lines 85-87) to communicate this issue more clearly.

Figure R1. Measurements of $[\text{Ca}^{2+}]_i$ in HEK 293 cells expressing hTRPV6-D542A mutant exposed to 0.2 or 2 mM extracellular Ca^{2+} in the absence or presence of 1 mM spermine. One representative measurement is shown from three independent experiments.

5. Tail Currents on figure 1: The tail currents recordings are very enlightening. The text states they “reach the maximum at +100 mV.” On one hand, the phrasing might be confusing for broad audience.; tail currents are usually measured at a fixed return potential after a prepulse. On the other hand, the figure shows them increasing with the prepulse voltage. This could be an indication of a relief from blocking or slow unblock kinetics. The total abolition of this phenotype in T539V (Fig 5) is a strong piece of evidence and should be discussed more in the main text as proof of the specific interaction site.

We apologise for the confusion and have changed our text to clarify our settings (lines 109-119). According to the step protocol shown in Fig. 1k, we applied voltage steps from -160 to +100 mV with a 20-mV increment and returned to a fixed potential (-100 mV). Applying this protocol to untreated cells (Fig. 1l) allowed us to detect large inward and relatively small outward TRPV6 currents, whereas tail currents were not detectable. In the presence of intracellular spermine (Fig. 1m), we observed (i) a significant reduction in inward and outward currents and (ii) development of pronounced tail currents. Fig. 1n shows the results of further analysis of tail currents from Fig. 1l-m. To visualise the effect of spermine better, we normalised tail currents at 201 ms (at -100 mV) to the averaged inward currents at -100 mV obtained from 0 to 90 ms. We observed that only after positive pre-pulse voltages, the application of spermine gradually increased tail currents, with a maximal response at +100 mV pre-pulse. We interpret these results as the TRPV6 block by spermine is voltage-dependent and can be relieved at positive potentials. The results for T539V and D580R in Fig. 5d-i were obtained using a similar approach except that (i) data were shown only for +100 mV pre-pulse in Fig. 5d-f and (ii) tail currents were not normalised in Fig. 5g-i. As requested, we have extended the description of our results for T539V (lines 250-260). Our experiments with T539V support the structural data and conclusions drawn from MD regarding the role of T539 in interaction with TRPV6.

6. Pose Stability in MD simulations: The MD data is used to bridge the gap between the static structure and function. “Pose 2” is described as transient. If it is transient, why does the EM density appear continuous through that region? If the EM density is an average, it implies high occupancy in both the SF and the central cavity simultaneously? Please discuss.

Cryo-EM density largely corresponds and coincides with Pose 3 (see Supplementary Fig. 8). Of course, some intracellular portion of this density can be contributed by Pose 2, which overlaps with Pose 3 at this location (see Figs. 4 and 6). Additionally, our MD analysis revealed that the transition time through Pose 2 depends strongly on ionic conditions. Thus, the relative occupancies of Pose 2 and Pose 3 in the cryo-EM data may differ from MD observations. We have added the corresponding discussion to the text of the manuscript (lines 193-227).

7. The proposed mechanism is “Stepwise.” Pose 1 (D580) to Pose 2 to Pose 3 (SF). The mutagenesis of D580R supports Pose 1 as an entry site. However, if D580R abolishes block, does it prevent entry or binding? The currents in D580R look large (Fig 5c), so the channel is functional. This could represent a control that validates the MD. For the D580R mutant, the authors show it is insensitive to spermine. It would be valuable to see if this mutant retains sensitivity to other blockers (like Ruthenium Red or cis-22) to prove the pore structure isn’t radically perturbed, only the polyamine entry path.

We agree with Reviewer 1 that additional control experiments with D580R-TRPV6 are beneficial for the study. Because Ruthenium Red and cis-22a interact with TRPV6 at a binding site that overlaps with that for spermine (PMID: 34725357, 33246965), we examined the effects of Gd^{3+} and econazole, which likely act through different mechanisms (PMID: 34725357, 21057859, 16356545, 10428857, 12869611, 11687570), and found that D580R-TRPV6 retains sensitivity to Gd^{3+} (new Supplementary Fig. 12) and econazole (new Supplementary Fig. 13). Additional MD simulations were performed for this mutant, and it was found that the spermine molecule did not penetrate the pore at all in three 1000-ns replicas (new Supplementary Fig. 14). Therefore, we suggest that arginine forms a potential barrier, which obstructs the spermine entry into the pore and thereby prevents the pore blockade. We have clarified these new findings in the text (lines 270-274).

Reviewer #2 (Remarks to the Author):

Summary:

The study by Neuberger et al. describes the cryo-EM structure of human TRPV6 captured in the presence of the polyamine spermine. Polyamines are ubiquitous and play critical roles in numerous essential life processes. They are dysregulated in many cancer types and, of particular relevance to this manuscript, broadly impact the function of numerous ion channels. How polyamines modulate the electrical character of these ion channels is unknown. In functional studies, the authors observe dose-dependent block of TRPV6 when spermine is administered from both the extracellular or intracellular side. Interestingly, intracellular spermine produced voltage-dependent tail currents in the presence of spermine, consistent with an open-channel block mechanism. The authors present a structure of human TRPV6 with spermine lodged in the selectivity filter (SF) of the channel pore. This is the major finding of the paper. MD simulations further spermine interacting with three intrapore locations, with one of the sites overlapping with the spermine-binding site observed in the cryo-EM structures. Mutagenesis and functional analysis confirm the importance of these sites for spermine-dependent channel modulation. Together, these findings support an open-channel block model in which spermine enters the pore from the cytosol to obstruct ion flow in a use-dependent manner.

We thank Reviewer #2 for the kind assessment of our work.

Major Comments:

1. While the overall cryo-EM reconstruction of the channel at 3.48 Å appears sound and in line with current standards, the evidence of the ligand pose is substantially weaker than for the surrounding protein scaffold. The cryo-EM density authors attribute to spermine is not well-defined, appearing fragmented and located on the four-fold symmetry axis, which raises concerns of over-interpretation. Spermine, as presented, is modelled asymmetrically with an internal “kink”, despite being situated in the symmetry axis of the SF and is not convincingly supported by the density or by clear specific interactions with surrounding residues. Given the somewhat fragmented density at this site, such fine-grained conformational detail is unlikely to be reliably resolved. The manuscript could benefit from more rigorous quantitative validation of the spermine model and from tempering statements that imply an unambiguous binding mode.

Reviewer #2 is right that the density for spermine is weaker than for the surrounding protein, and this is likely due to the dynamic and not very potent nature of spermine binding. In addition, the blocker in the pore has the freedom to rotate around the channel axis of symmetry, which also blurs the corresponding density in the cryo-EM map. We have seen these weakening of density even for high-affinity ion channel blockers in AMPA and kainate receptors (doi: 10.1016/j.neuron.2018.07.027; doi: 10.1038/s41467-024-54538-x). The corresponding explanations have been added to the text (lines 149-154).

2. The MD simulations demonstrate that spermine docks at three intrapore poses and with greatest stability at the SF (pose 3) close to the position taken up by spermine in the cryo-EM structure obtained in the absence of Ca²⁺. While this correlation is intriguing, this does not by itself establish that this is the predominant functional blocking site. Additionally, simulations with Ca²⁺ present indicate that Ca²⁺ occupancy of the SF prevents spermine docking to the SF from the intracellular side. This finding appears to argue against pose 3 as the dominant blocking configuration under physiological conditions where Ca²⁺ is expected to permeate the channel. This important constraint is not discussed. Furthermore, it is unclear whether Ca²⁺ flow can be blocked if spermine were to occupy only poses 1 and/or 2. It would strengthen the manuscript to clearly distinguish between what is directly supported by the structural and MD data (i.e. a plausible binding region and plausible blocking configurations under specific access and ionic conditions) versus what is hypothetical. The latter should be explicitly framed as a working model rather than as a definitive mechanism.

We agree that our simulation setups only approximately reflect the physiological ionic conditions. Furthermore, in our recent work, we demonstrated the presence of several Ca²⁺ binding sites along the TRPV6 pore (doi: 10.1016/j.str.2024.10.018). However, we believe that the site in the selectivity filter is the most energetically preferable for Ca²⁺ because of the strongest negative electrostatic potential induced by D542 carboxylic groups in the pore. This site is also the only one occupied by Ca²⁺ in many cryo-EM maps of TRPV6 (e.g., see the aforementioned work for more details), which further suggests its highest affinity. So, we propose that spermine-Ca²⁺ competition for this site is an effective test of relative affinity. Additional Ca²⁺ in the other pore sites would slow down the spermine penetration into the filter, but would not alter it.

Indeed, in setup 2, spermine did not knock off Ca²⁺ from the selectivity filter. We suggest that the cause is the kinetic barrier that prevents spermine binding during the times of our simulations. In particular, the presence of Na⁺ ions in the pore central cavity between spermine and Ca²⁺ hinders the rapid spermine penetration. Their escaping from the pore is a much slower process in the presence of Ca²⁺ than in divalent-free setups because Ca²⁺ binding to the SF is stronger than the binding of Na⁺. To clarify this, we performed simulations with setup 3, in which spermine starts from the pore (Pose 2) without Na⁺ ions in the cavity. In this configuration, spermine can compete directly with Ca²⁺ for the filter site. As can be seen in Supplementary Fig. 7, the spermine quickly knocks off Ca²⁺ and adopts Pose 3 in all three MD replicas. These results support the idea that the same stepwise blocking mechanism occurs more slowly in the presence of Ca²⁺ compared to the divalent-free conditions. Further support of this mechanism is provided by the additionally performed patch-clamp experiments, which show that Ca²⁺ currents are inhibited in a similar way as

monovalent ion currents under the application of extracellular/intracellular spermine (Fig. 1, new Supplementary Figs. 1 and 2).

In the other poses, spermine also occludes the pore and would prevent ion current. However, we suggest that these configurations have a minor contribution to the pore blockade due to their lower stability and affinity compared to Pose 3 (see also our other responses below).

3. The MD simulations are used as key support for the proposed inhibition mechanism but the scope is more restricted than the text implies. While the simulations demonstrate the stability of intracellular spermine docking to pose 3 in the SF under strong constraints, they do not establish that spermine can access the pore from the extracellular side or that the full inhibition pathway has been captured. The manuscript does not clearly identify which pose is proposed to mediate block under physiological Ca^{2+} with extracellular spermine versus intracellular spermine in Ca^{2+} -free conditions.

We agree with Reviewer 3 that spermine may act through multiple mechanisms on TRPV6, and we have communicated this idea in the revised manuscript (lines 319-324). However, we would like to emphasize that our study focuses on a specific **structure-driven** model, complemented by MD simulations and electrophysiological assessment of WT and mutant variants of TRPV6, rather than relying on a single approach. Specifically, our cryo-EM data showed no non-protein density in the extracellular side of the TRPV6 selectivity filter. Consistent with this, the new MD simulations with extracellular spermine did not reveal its penetration into the filter. Instead, spermine displayed a highly dynamic pose within the filter vestibule (new Supplementary Fig. 8k, setup 5). Based on this, we do not foresee that the current data provide solid evidence for a direct inhibitory effect of extracellular spermine on TRPV6, and thus, we do not explore this mechanism in the present study.

At the same time, the intracellular stepwise block and Pose 3 as the main site are strongly supported by the following observations. (i) Cryo-EM data only visualised non-protein densities in the pore corresponding to Pose 1 and Pose 3. (ii) Stepwise binding to Pose 3 observed in MD simulations under various conditions: only Na^+ in the pore (setup 1), Ca^{2+} in the selectivity filter (SF) (setup 3), a second spermine molecule in the extracellular vestibule of the SF (setup 5). (iii) Mutations of residues involved in the intracellular spermine binding alter the TRPV6 inhibition in the patch-clamp experiments. These numerous observations provided by different methods seem to be robust enough to support the proposed mechanism.

4. The functional evidence for block is distributed across assays that probe different conditions and access route. Calcium imaging demonstrates reduced Ca^{2+} influx when spermine is applied extracellularly under physiological Ca^{2+} , whereas whole-cell recordings are performed in nominally Ca^{2+} -free conditions with spermine in the pipette solution. These regimes are not equivalent and make it difficult to extract a single, unified blocking mechanism. With the electrophysiological data, the appearance of prominent, voltage-dependent tail currents only in the presence of the ligand, despite stable, non-inactivating

currents during the depolarizing prepulse and minimal tails in the control, suggests that spermine not only blocks the pore but also alters deactivation gating and/or stabilizes an additional kinetic state. The current text focuses on voltage-dependent open-channel block and does not discuss the likely impact on deactivation kinetics or the state-dependence of block under the different ionic and spermine-application conditions. A more balanced interpretation that explicitly considers spermine-induced changes in deactivation would provide a more accurate structure-function picture and prevent over-simplification of the mechanism.

Thank you for highlighting this important point. Ca^{2+} imaging and patch-clamp electrophysiology were used as complementary techniques because these methods operate under very different experimental conditions, thereby providing stronger evidence for the inhibitory effect of spermine on TRPV6. Specifically, Ca^{2+} imaging allowed evaluation of TRPV6 activity in resting cells and during extracellular ligand application. Please note that in mammalian cells, extracellular spermine is thought to enter via poorly understood polyamine uptake systems. There is evidence that endocytic and solute-carrier-type mechanisms can contribute to such uptake (PMID: 20522643, 21318884, 12972423, 29408503). In whole-cell recordings, precise control of membrane voltage and intracellular solution composition was enabled, with TRPV6 exposed to cytosolic spermine, mimicking its primary physiological location. Hence, combining these approaches was not intended to create identical regimes but to offer less biased insights into the channel's response to spermine.

Nevertheless, we agree with Reviewer #2 that the inhibitory effects of spermine deserve further insights. Therefore, we conducted additional patch-clamp experiments to assess the effect of externally applied 1 mM spermine on Na^+ and Ca^{2+} TRPV6 currents (new Supplementary Figs. 1 and 2) and found that TRPV6 remains sensitive to spermine, although the inhibitory effect was less pronounced compared to the application of 1 mM intracellular spermine (Fig. 1e), supporting the idea that spermine enters the channel pore from the cytosolic side. In addition, we examined the effect of 1 mM intracellular spermine in new settings, allowing us to monitor Ca^{2+} TRPV6 currents (new panels h-j in Fig. 1). This approach also revealed a significant inhibition of TRPV6 by intracellular spermine. We also clarified our text related to the tail currents obtained with the step protocol (see our response to point 5 of Reviewer #1 above).

Reviewer #2 also noted that the presence of spermine caused TRPV6 to exhibit “stable, non-inactivating currents during the depolarizing pre-pulse” and suggested that this might indicate altered activation/inactivation kinetics or a state-dependent effect of spermine. Unfortunately, this appears to be a misunderstanding. In the presence of spermine, outward currents increased by up to twofold during depolarizing pre-pulses above +60 mV (**Figure R2**), whereas currents in the corresponding control recordings remained stable. These findings most likely reflect relief of spermine block at positive potentials, or even displacement of spermine from the channel pore. Notably, the kinetics of the current increase during the positive pre-pulse were comparable to those of the current decrease upon return to the holding potential of -100 mV (Fig. 1m). Altogether, our analysis of the tail currents supports voltage-dependent unbinding and re-block by spermine in the ion channel pore. Moreover, our structural data did not reveal any conformational changes or alternative states associated with spermine binding in the open pore.

Figure R2. Close-up view of the outward currents shown in Fig. 1l,m. Representative measurements are shown for $n = 6-10$ cells examined. Whole-cell Na^+ currents recorded in the absence (black) or presence (blue) of 1 mM intracellular spermine using the voltage step protocol shown in Fig. 1k. Currents are shown for the pre-pulse potentials in the range from 0 to +100 mV with 20 mV increments.

Finally, we have revised the text (lines 319-329) to offer a more balanced view of the inhibitory effects of spermine. Specifically, we now highlight that our study concentrates on one particular mechanism driven by structural results rather than examining all possible effects of spermine. Since spermine is a polyvalent cation, it can interact with negatively charged membrane lipids, thereby impacting plasma membrane properties such as surface charge, lipid organisation, and membrane stability, which can influence deactivation/inactivation kinetics of TRPV6, and affect the channel's state differently.

Reviewer #3 (Remarks to the Author):

The manuscript by Neuberger et al. reports on an interesting investigation of the inhibition mechanism of TRPV6 by spermidine. The authors use a combination of techniques including cryoEM, molecular dynamics, and electrophysiology to clarify the molecular underpinnings of the regulation of human TRPV6 by polyamine spermine. The paper sheds light on one of the fundamental mechanisms of modulation of ion channels and provides novel structural information about the binding of polyamines to TRPV6. The results are of sufficiently wide interest to warrant publication. I find that the simulations are rigorously carried out and that the manuscript is, for the most part, clearly written and easy to read. I recommend publication after the following concerns have been addressed:

We thank Reviewer #3 for the generous assessment of our work.

1) While I find the information provided by the molecular dynamics simulations insightful, one is left wondering whether different replicas would end up resulting in the same poses as those described in the paper. I think that a potential of mean force for the binding/unbinding of the polyamine would make the arguments discussed in the paper more solid and reliable.

Indeed, in some of our MD-replicas, spermine was found to be trapped in the pore and could not penetrate in the main binding Pose 3. This is especially striking in Setup 1, where the stepwise blocking mechanism can be clearly seen in most of the replicas. We performed two

additional independent replicas with this setup (1.5 us in total, new Supplementary Fig. 6), and in both cases, spermine adopted Pose 3, which enhanced the statistical significance of our conclusions. To strengthen the interpretation of the experimental results, we also performed multiple replica calculations with two spermine molecules starting simultaneously from the extracellular and intracellular sides (Setup 5, new Supplementary Fig. 8), and with the TRPV6 mutant forms T539V (new Supplementary Fig. 14a-g) and D580R (new Supplementary Fig. 14h-j). In total, 14.5 us of additional MD-trajectories were calculated.

PMF calculation is a really powerful instrument for estimating binding affinity and kinetic barriers. We employed these methods in our previous studies (e.g., see DOI: 10.1038/s41467-025-65919-1). We also tried to calculate the PMF along the spermine stepwise binding path, but encountered a convergence problem. The following issues arise when using Umbrella sampling or Metadynamics-like methods to calculate such PMF. (i) Spermine motion along the pore is accompanied by the cations knocking off via a narrow selectivity filter. This process is quite slow, leading to a hysteresis problem – the PMFs calculated along the binding and unbinding paths do not converge. (ii) Spermine is a flexible molecule, and its full conformational ensemble has to be sampled for the reliable free energy estimation, especially in the highly dynamic Pose 1. At the same time, spermine flexibility should be “frozen” to provide its motion along the pore under the action of the Metadynamics repulsion potential. For these reasons, achieving a converged PMF would require the selection of some sophisticated collective variables (CVs) and extensive calculations. However, it should be noted that the objective of this work was not to strictly quantify the free energy landscape of the spermine in the pore, but to delineate the most probable mechanism of channel blockade. As it turned out, to solve this problem, it was enough to collect good statistics using a large series of independent MD simulations and considering various scenarios of spermine behaviour in the pore. Direct comparison of the calculation results with the experimental data was also of key importance. For this reason, after careful consideration, we decided not to perform PMF calculations.

2) The calculations used to produce the plots in figs S6 and S7 (ESP and MHP) should be described in the methods. At the moment, the manuscript refers to a webserver, which, in turn, cites the relevant paper. A concise summary of the methodology should be reported in the manuscript.

We have added the requested details to the Methods section (lines 525-538).

We are very thankful to the Reviewers for their time and effort. We have addressed the additional comments of Reviewer #2, with the details outlined below.

I appreciate the revision and the additional experiments. The manuscript is improved and the new data strengthen the conclusion that spermine inhibits TRPV6. My remaining concerns are mainly about mechanistic interpretation.

We are pleased that Reviewer #2 found that the additional data in the revised manuscript strengthened our conclusions. In response to the reviewer's suggestion, we made additional changes to the text to better balance our interpretation of the results, as outlined below.

1. My original comment on the tail-current phenotype appears to have been misread. The phrase "stable, non-inactivating currents during the depolarizing prepulse and minimal tails in the control" referred to the control condition, not to spermine. My point was that the ligand-dependent emergence of strong tail currents introduces kinetic complexity that deserves balanced discussion.
2. I remain unconvinced that the tail currents uniquely support a simple interpretation of relief of spermine block at positive membrane potentials. Although the tails are consistent with voltage-dependent changes in spermine-channel interactions, the authors' own response emphasizes that the prepulse currents in the presence of spermine remain strongly suppressed at positive potentials. The current text should be toned down to state that the phenotype is consistent with, but not uniquely diagnostic of voltage-dependent relief of block and that more complex state-dependent kinetics are not excluded.

We are sorry for the misunderstanding of the raised point. As suggested, we introduced the following statement in the revised text (lines 120-122 and 302-303): The observed effects of spermine on tail-current appearance are consistent with voltage-dependent relief of spermine block, although they are not uniquely diagnostic of this mechanism, and more complex state-dependent kinetics remain possible²¹.

Moreover, our revised Discussion statement contains an additional statement that the spermine can regulate TRPV6 through additional mechanisms (lines 328-333): It is important to emphasize that our study focuses on a particular structure-driven model and, therefore, does not exclude the possibility of alternative mechanisms of hTRPV6 regulation by spermine. As a polyvalent cation, spermine can interact with negatively charged membrane lipids and thereby alter the plasma membrane properties, including surface charge and membrane stability⁷⁶. Such effects could, in turn, modulate the functional properties of TRPV6 through mechanisms distinct from the one proposed here.

3. The explanation offered for extracellular spermine effects, namely uptake into cells and indirect access to the intracellular blocking pool, remains speculative in the context of the present study. If retained, this point should be either supported directly or clearly identified as speculative. The new external patch-clamp data are helpful but do not fully resolve this issue.
4. Patch clamp experiments establishing the concentration dependence of intracellular spermine block would still help interpret the calcium imaging and electrophysiology results.

As suggested, we revised the text to clarify that, although the effects of extracellular spermine on TRPV6 may be explained by cellular uptake, additional studies are needed to verify this model (lines 296-299). We now also refer to a review summarizing numerous

studies showing cellular accumulation of extracellularly applied radiolabelled putrescine, spermidine, and spermine (PMID: 8811834), along with another original study based on fluorescent polyamine uptake imaging (PMID: 15208319). In this context, we do not believe that repeating these classical experiments is necessary to demonstrate spermine uptake under our specific conditions. At the same time, we do not think that the concentration dependence of intracellular spermine block of TRPV6 currents provides a reliable basis for either proving or ruling out this option. Please note also that the revised manuscript contains a considerable body of new electrophysiological data to show the inhibitory effect of externally applied spermine on Na⁺ and Ca²⁺ TRPV6 currents (new Supplementary Figs. 1 and 2), although the inhibitory effect was less pronounced compared to the application of intracellular spermine (Fig. 1e). In addition, we examined the effect of intracellular spermine in new settings, allowing us to monitor Ca²⁺ TRPV6 currents (new panels h-j in Fig. 1). This approach also revealed a significant inhibition of TRPV6 by intracellular spermine, consistent with the idea that spermine enters the channel pore from the cytosolic side.